# HAMLET: A Hierarchical and Adaptive Multi-Agent Framework for Live Embodied Theatrics

**Shufan Jiang**[†∗1,4]  **Sizhou Chen**[∗2,3,4]  **Chi Zhang**[3]  **Xiao-Lei Zhang**[✉3]  **Xuelong Li**[✉3]

[1]East China University of Science and Technology  [2]The University of Sydney
[3]Institute of Artificial Intelligence (TeleAI), China Telecom  [4]Datawhale Org.

## Abstract

Creating an immersive and interactive theatrical experience is a long-term goal in the field of interactive narrative. The emergence of large language models (LLMs) provides a new path to achieve this goal. However, existing LLM-based drama generation methods often produce models that lack initiative and cannot interact with the physical scene, while typically requiring detailed user input that diminishes the immersion of live performance. To address these challenges, we propose HAMLET, a hierarchical adaptive multi-agent framework focused on drama creation and real-time online performance. Given a simple topic, the framework first generates a narrative blueprint to guide the subsequent improvisational performance. In the online performance phase, each actor is equipped with an adaptive reasoning module that enables decision-making based on their personas, memories, goals, and emotional states during complex group chat scenarios. Beyond dialogue, actor agents engage in embodied interactions by changing the state of scene props through actions such as opening a letter or picking up a weapon, which are broadcast to update the global environmental context. To objectively assess the quality of live embodied theatrics, we establish a comprehensive evaluation method and introduce HAMLETJudge, a specialized critic model for automated evaluation. Experimental results demonstrate that HAMLET excels in creating expressive, coherent, and physically interactive theatrical experiences in an autonomous manner.

## 1 Introduction

In recent years, LLMs have demonstrated strong ability in various generative tasks, especially in creative fields such as story creation (Li et al., 2024) and role-playing (Tu et al., 2024). These models are able to generate fluent and imaginative texts, providing a foundation for the development of interactive narrative. In applying these capabilities to drama creation, research efforts can be categorized into two main directions: drama generation and drama performance.

Existing studies have explored drama generation employing hierarchical approaches (Fan et al., 2018; Yao et al., 2019) and modular script structuring (Mirowski et al., 2023), yet significant challenges remain in applying LLMs to structured drama generation, especially for online real-time enactment. In interactive drama performance, LLM agents take on the role of performing the narrative, where, in contrast to free-form story creation, each actor must make decisions and take actions consistent with their profiles and the overall plot progression. These interactions must unfold across a series of scenes to collaboratively advance the narrative. Furthermore, existing methods typically require detailed user input, such as full story outlines (Wu et al., 2024) or elaborate guiding paragraphs (Wu et al., 2025b), which not only increases design costs but also limits storyline flexibility.

---

∗Equal Contribution. †Project Lead. ✉Corresponding Author. *Correspond to xiaolei.zhang@nwpu.edu.cn and xuelong_li@ieee.org*

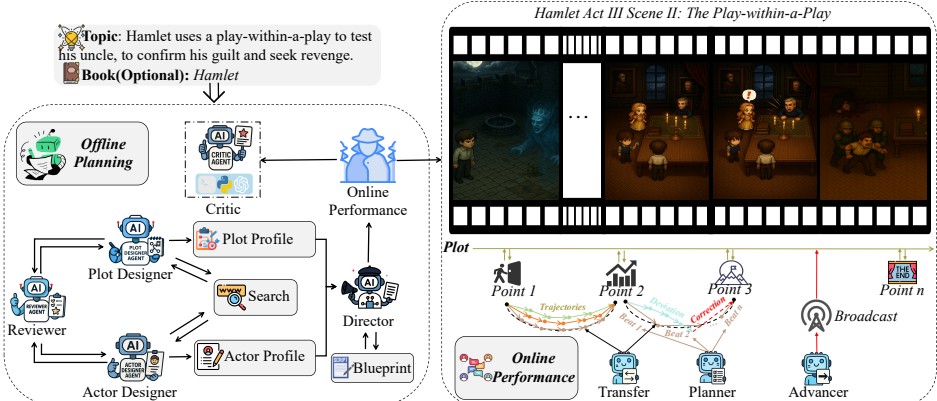

Figure 1: HAMLET creates AI drama in two main stages. First, during offline planning, a collaborative workflow of agents including the actor designer, plot designer, and reviewer creates initial materials, which are then integrated by a director agent into a structured narrative blueprint. The blueprint then guides the subsequent online performance, where a control system composed of a planner, transfer, and advancer directs a dynamic and improvisational theatrical experience.

A key challenge in drama performance lies in redefining the interaction patterns of LLM agents. In many prior studies, these patterns are limited to passive, turn-based responses. For example, some studies focus on the adequacy of single-turn dialogues rather than active participation (Wu et al., 2024). The agents typically await user instructions, lacking genuine proactivity. This reactive model struggles to support dynamic scenes where multiple AI actors interact with human players. We believe that truly vibrant, dramatic interactions require AI actors to make autonomous decisions, collaborate or conflict in open scenarios, and actively guide the development of the plot. This paradigm shift from passive response to active guidance provides a new perspective on how to enhance actor initiative in dramatic performance.

Beyond autonomous agent interaction, a comprehensive framework for drama performance requires two essential components, physical environment interaction and a holistic evaluation system. Drama is inherently an art form combining language with embodied action, where actor agents' behaviors dynamically influence their surroundings while environmental feedback in turn plays a crucial role in shaping performance. Without this embodied dimension, drama risks degenerating into abstract dialog lacking the tangible presence and realism that define compelling theatrical experience. Moreover, no established method effectively evaluates real-time drama quality, as existing benchmarks predominantly assess textual coherence (Gómez-Rodríguez & Williams, 2023) or role-playing fidelity (Tu et al., 2024; Wu et al., 2025a; Wang et al., 2025), rather than the integrated impact of complete dramatic enactment.

To address the above challenges, we propose HAMLET, a multi-agent framework that enables **H**ierarchical and **A**daptive **M**odeling for **L**ive **E**mbodied **T**heatrics. Specifically, we make the following contributions:

1. HAMLET: Our multi-agent framework comprises two decoupled stages of offline planning and online performance. The offline stage requires only a simple topic to generate a structured narrative blueprint, providing freedom for improvisation during online performance while ensuring the integrity of the main story structure. To enhance online performance, each agent is equipped with a **P**erceive **A**nd **D**ecide (PAD) module that acts as a pre-response mechanism, making their perceptions and decisions more human-like.
2. Comprehensive evaluation: To objectively assess the quality of drama generation and performance, we establish a comprehensive evaluation method that assesses three curated dimensions. This method incorporates a leaderboard that uses GPT-4o as a strong baseline for win rate comparisons. Additionally, we trained HAMLETJudge, a critic model designed for cost-effective and reliable drama performance evaluation.
3. Extensive experiments: We conducted evaluations and detailed ablation studies on HAMLET's core components. Notably, both PAD and HAMLETJudge, are 8B models, yet they achieved state-of-the-art performance in their tasks respectively.

## 2 RELATED WORK

**LLM-Based Drama.** Research on LLM-based drama has evolved along two main directions: drama generation and drama performance. Initial efforts in drama generation adapted techniques from general story creation, using hierarchical models to plan plots and generate coherent narratives (Fan et al., 2018; Yao et al., 2019). With the development of this field, researchers have tried multi-LLM collaboration (Venkatraman et al., 2024; Mirowski et al., 2023; Han et al., 2024) that mainly focus on drama scripts generation. A key limitation of previous work is that they generally require a complete story (Wu et al., 2024) or a leading paragraph (Wu et al., 2025b) as input. We present a framework that supports arbitrary and customizable drama generation. At the same time, research on role-playing in drama performance is predominantly conducted through LLM-based agents (Chen et al., 2024b). These agents typically require detailed knowledge about the actor in order to model their expressive style (Shanahan et al., 2023; Wang et al., 2023; Tu et al., 2024). Existing role-playing systems are generally built using either prompt engineering approaches (Li et al., 2023; Chen et al., 2023) or fine-tuning methods (Shao et al., 2023; Lu et al., 2024). However, these techniques struggle to capture rich and nuanced character traits (Huang et al., 2023), and consequently fail to produce truly live and fully embodied dramatic performances.

**Evaluation of Role-Playing Conversation Agents.** Traditional dialogue metrics like Embedding Similarity, BLEU, and ROUGE (Mou et al., 2016; Wu et al., 2020; Serban et al., 2017) are widely used in role-playing conversation agent (RPCA) research (Wang et al., 2024; Zhou et al., 2024; Tao et al., 2024). However, these quantitative indicators struggle to evaluate role consistency and personality. To address this, specialized evaluation frameworks have been proposed. RoleEval (Shen et al., 2023) uses role-specific multiple-choice questions to test the model's understanding of a character. SocialBench (Chen et al., 2024a) constructs evaluation questions from multi-source dialogues, while CharacterEval (Tu et al., 2024) adopts multi-round dialogues and multi-dimensional scoring for aspects like conversational ability. Additionally, RAIDEN (Wu et al., 2025a) builds a Q&A dataset via annotator interaction to evaluate responsiveness in specific dimensions. Although these methods provide quantitative criteria, they are typically limited to two-character or user-character scenarios. While CoSER (Wang et al., 2025) expands the number of roles, it still lacks an evaluation mechanism for overall dramatic performance. Motivated by these limitations, we propose new criteria to evaluate the complete dramatic performance.

## 3 HAMLET

To achieve an automated, interactive, and expressive AI drama experience, we designed HAMLET, as shown in Figure 1. The framework decouples the generation and performance of the drama into two main stages: offline planning, a multi-agent collaborative workflow to generate a narrative blueprint, and online performance, a hierarchical control system that executes the blueprint, manages real-time interactions, and handles environmental feedback.

### 3.1 OFFLINE PLANNING

The objective of the offline planning stage is to transform user input into a structured narrative blueprint. This stage is designed to handle two types of input: i) An arbitrary and customizable topic. For this input type, the offline workflow is able to generate a complete act based on the provided topic. ii) A complete literary work. When given a full literary text, the workflow first deconstructs it into a series of acts based on its chapters and content structure. Subsequently, it proceeds with drama design for each act, ensuring the adaptation remains faithful to the source material.

**Agent-Based Workflow Architecture.** The offline workflow consists of four agents: the actor designer, the plot designer, the reviewer, and the director. This workflow implements character creation, plot generation, review, and final integration.

**Character Profile Generation and Review.** The starting point of the workflow is actor construction. The actor designer is responsible for generating actor profiles of core characters based on user input. To ensure the depth and accuracy of character creation, this agent queries an external knowledge base via its search module. The output is a structured profile that defines the static attributes of the character, such as background and personality, and dynamic attributes, such as their initial goal

and core relationships within the story. The profile is then submitted to the reviewer, who checks the rationality of the character settings, the clarity of motivations, and the relationships between actors.

**Plot Generation and Structuring.** After all actor profiles are approved, the plot designer composes a preliminary narrative draft based on the topic and actors. This draft is then submitted to the reviewer for evaluation. Once the review is passed, the director is responsible for the final structural processing, reconstructing the linear story draft into a hierarchical plot profile. This process includes the following key steps: 1) Act and scene definition: Divide the drama into several acts and specify the scenes in which each act takes place. 2) Creation of environmental elements: Generate a list of interactive props for each scene. This list has specific descriptions and location information for the props. 3) Narrative points: Define a series of narrative points within each act. Each point contains a clear flag and a result that marks its completion. 4) Backward planning: To prevent plot deviation, the director will give priority to generating the end point when planning an act. Then, based on this end point, it will supplement and construct the logically coherent preceding points leading to the ending in a reverse manner.

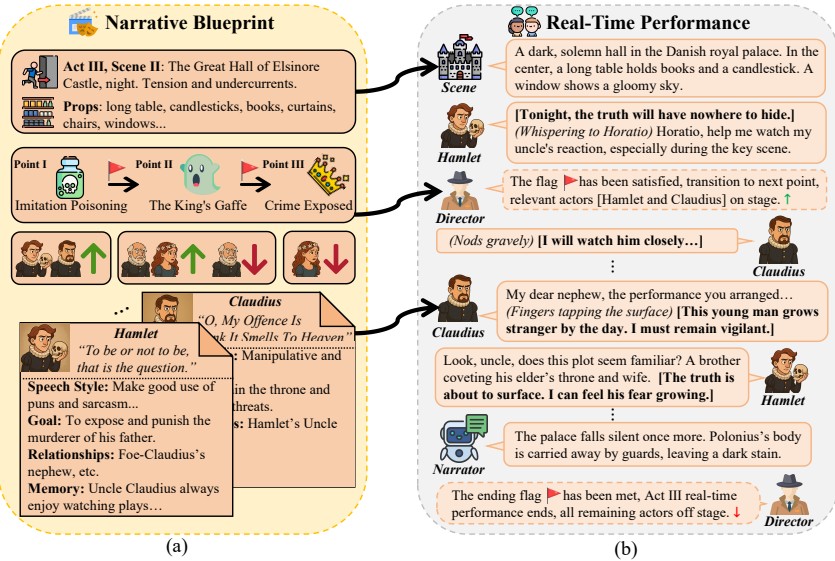

Figure 2: An illustration of HAMLET's core components for performance generation: a narrative blueprint that defines the scene, plot and character profiles, and the resulting real-time conversation containing scene descriptions, dialogue and narrator judgment.

Finally, the director integrates the plot profile with the actor profile to generate a narrative blueprint which is shown in Figure 2(a) as the input of the online performance stage.

## 3.2 ONLINE PERFORMANCE

Upon receiving the narrative blueprint, the online performance stage begins to transform it from a static plan into a dynamic, interactive, and immersive environment. This environment will accommodate both autonomous AI actors and human players who assume certain roles. To ensure the performance quality, this phase introduces more specific narrative units, environmental interaction mechanisms, and a group of collaborative agents. An example of online real-time performance is illustrated in Figure 2(b).

**Performing Drama.** Online performances are organized into acts, each comprising multiple scenes and narrative points. Scenes define the physical environment and interactive props, establishing the spatial context, while points specify plot objectives and serve as milestones for narrative progression. Together, these elements structure the complete dramatic process within an act.

The narrative path between two points is formed by a dynamically generated trajectory which consists of a series of beats. We define a beat as a meaningful interaction step where an actor performs an effective action. This action logic is driven by a dual-goal system. For each beat, an actor's deci-

sion references both the public flag of the current point and a personal private goal that is refreshed at the beginning of every act.

Due to the actors' autonomy, multiple trajectories can connect $point_i$ to $point_{i+1}$, such as one composed of $beat_{j_1}, beat_{j_2}, \ldots, beat_{j_n}$ and another of $beat_{k_1}, beat_{k_2}, \ldots, beat_{k_m}$. These multiple combinations introduce a high degree of freedom and arbitrariness to the performance.

**Environment Interaction.** Credible physical interaction is the core challenge of the online performance stage. As shown in Figure 3, we designed a narrator agent to adjudicate all interactions between actors and the environment. The role of the narrator is to ensure the rationality of all physical actions. When an actor attempts to perform a physical action, the narrator will make a judgment based on the environment state and physical rules. If the action is feasible, the narrator will confirm its success, update the environment state, and broadcast an objective description to all participants. Otherwise, the narrator will determine failure with a logical explanation.

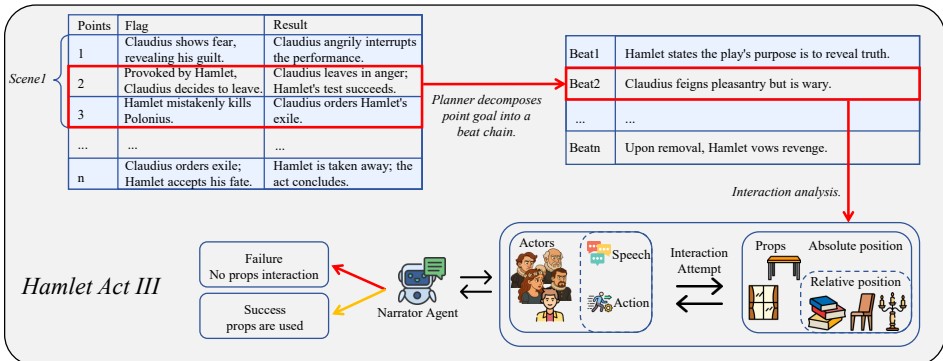

Figure 3: An example of the real-time interaction and adjudication loop in the online performance. An actor agent attempts an action or speech, termed a beat, to progress towards the current narrative point. The narrator agent then intercepts this attempt, determines whether it is a success or failure, and provides objective feedback to all participants in the drama.

## 3.3 PERCEIVE AND DECIDE MODULE

**Hierarchical Control of Performance Effects.** All actor agents in the drama performance utilize a layered architecture, which is composed of an LLM and a PAD module. The LLM is responsible for generating specific speech and actions, while PAD handles the strategic decision-making that guides them. The design of PAD is detailed in the following subsection.

To ensure the online performance balances improvisational freedom with stable progression along the main storyline, we designed three collaborative control agents: 1) Planner pre-designs and reviews multi-trajectory results. It breaks down the flag into a few sequences of executable beats which form a trajectory. Beats can be references for advancer to guide the plot forward. 2) Transfer moves the story to the next point by regularly checking if the flag is met. Once satisfied, it advances the drama to the next point and manages relevant actors to enter or leave. 3) Advancer ensures the plot progresses. If it stalls beyond a set time threshold, the advancer directs necessary actors based on the current flag or next beat. Their relationship is illustrated in Figure 3.

In HAMLET, the PAD module plays a crucial role in guiding the actor agents to generate final responses. The design of PAD is conceptually grounded in the dual-process theory of human cognition (Kahneman, 2011). According to this theory, human thinking is characterized by two distinct modes of processing: System I, which is fast, automatic, and intuitive; and System II, which is slower, deliberate, and analytical. PAD integrates both intuitive and reflective reasoning mechanisms to better simulate human-like decision-making in complex and nuanced drama contexts. Specifically, PAD is mainly responsible for generating a decision of fast, slow, silence or potential actions by tool calls—to simulate and extend the dual-system mechanism. This expanded set of response modes enhances the module's compatibility with the interactive real-time dramatic context proposed in the study.

**Design Principles.** As depicted in Figure 4, the input of PAD is based on a dual principle. When perceiving, PAD acts as an actor possessing both subjective awareness of its internal state and an objective perception of external stimulus. When a new event occurs, PAD integrates information from both perspectives to ensure well-grounded decision-making.

The internal state, which represents the actor's self-awareness, is maintained as a live profile comprising both static and dynamic components. The static components include *Persona* and *Subjective Relationships*, while the dynamic elements, such as *Goal* and *Memory*, are retrieved or updated as needed. Upon the conclusion of an act or scene, key events and interactions are distilled into memory entries. These entries are stored and later retrieved via Retrieval-Augmented Generation (Lewis et al., 2020) to inform future behavior. This memory compression mechanism prevents exponential context growth, thereby enabling scalable support for long-form drama performance.

On the other hand, external stimulus constitutes objective environmental and contextual information. The external stimulus is composed of *Environment Description*, *Actor List*, *Dialogue History*, and *Interactable Objects*. Together, these components form a comprehensive representation of the external context, facilitating the integration of situational awareness into the decision-making process.

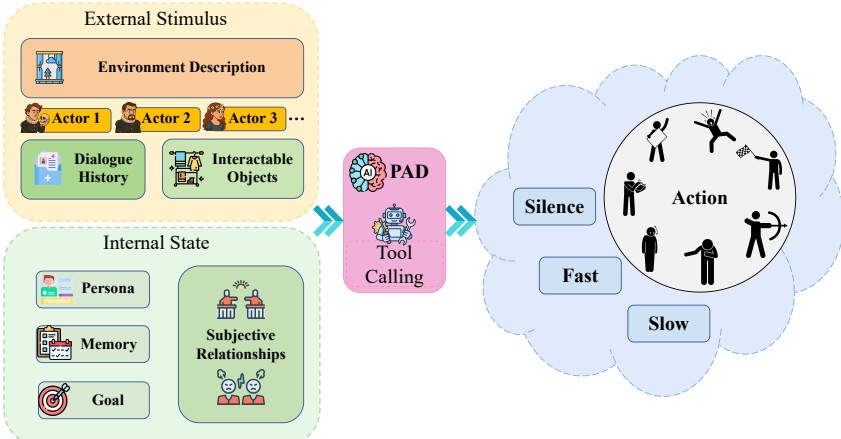

Figure 4: PAD processes external stimulus and internal state to determine a response strategy by tool calling.

**Decision-making Process.** Abstract context is translated into concrete speech and executable actions through a two-stage pipeline. As shown in Algorithm 1, PAD first selects a high-level response strategy that determines both the timing of the agent's reaction and a structured action, which is parsed into an executable triplet. Optionally, it also generates the underlying reasoning for this decision in the form of an internal monologue. In the second stage, the selected strategy, the parsed action, and the optional internal monologue jointly guide the actor agent in producing its final, concrete behavior. A detailed version of the algorithm can be found in Appendix D.

---

**Algorithm 1** Working Principle for PAD Module

---

**Require:** Actor $a_k$, Dramatic Context $\mathcal{C}_{drama}$, PAD Model Parameters $\pi_\theta$
**Ensure:** A final decision tuple $(r, o)$, where $o$ is the structured action (composed of subject, verb, and object) or empty set $(\varnothing)$, $r \in \{\text{FAST}, \text{SLOW}, \text{SILENCE}\}$ is the response strategy.
1: **procedure** GETACTORDECISION($a_k, \mathcal{C}_{drama}, \pi_\theta$)
2:    $prompt_{pad} \leftarrow$ EncodeStrategy($a_k, \mathcal{C}_{drama}$)
3:    $\hat{r}, [\hat{thinking}] \leftarrow$ PAD($prompt_{pad}, \pi_\theta$)
4:    $r \leftarrow \arg\max_{r' \in \{\dots\}} P(r'|\hat{r})$
5:    $\hat{o} \leftarrow$ GenerateAction($r, [\hat{thinking}], a_k, \mathcal{C}_{drama}$)
6:    $o \leftarrow$ ParseAction($\hat{o}$)
7:    **return** $(r, o)$
8: **end procedure**

---

## 4 EVALUATION METHOD

**Dataset Construction.** The evaluation dataset was constructed from 100 diverse cases, comprising 50 literary excerpts and 50 custom-authored drama topics. The literary excerpts are selections from 25 classical Chinese[1] and 25 renowned English[2] literary works. The 50 custom topics, designed to span 10 distinct themes, were developed and reviewed carefully by annotators. More details about the dataset can be found in Appendix A. Figure 5 illustrates the thematic distribution of the entire dataset.

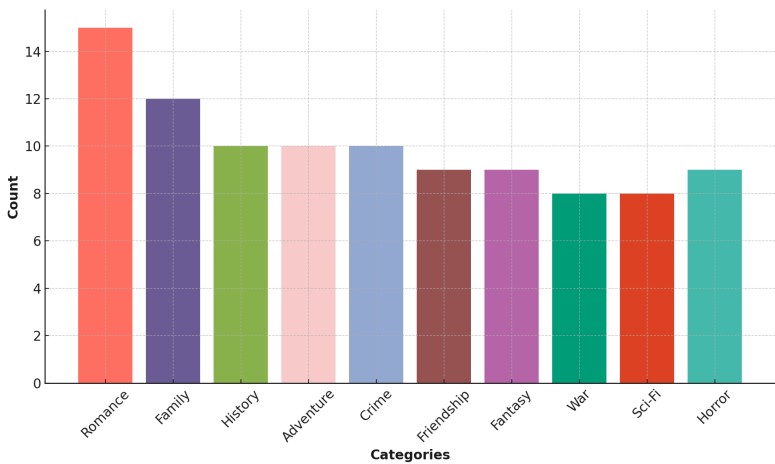

Figure 5: Distribution of drama topics across the dataset.

**Dimension Definition.** Defining evaluation dimensions is crucial for quantifying real-time drama performance. We categorize our evaluation into three core dimensions as follows:

1. **Character Performance (CP)** measures the quality of AI actors. This dimension evaluates *Believability*, i.e., the consistency of characters with their established personas, and *Agency*, which reflects the richness of emotional expression and the character's ability to effectively advance the narrative.
2. **Narrative Quality (NQ)** examines the overall craftsmanship of the story. This includes its *Coherence*, ensuring the logical development of the plot; its *Resonance*, measured by its thematic relevance and depth; and its *Integrity*, which evaluates the structural completeness of the storyline from beginning to end.
3. **Interaction Experience (IE)** focuses on the engagement of AI actors with the system. This dimension encompasses the quality and timeliness of the system's reactions, termed *Responsiveness*; the level of cognitive and emotional engagement, or *Immersion*; and the overall technical smoothness of the interaction, referred to as *Fluency*.

**Evaluation Principle.** Our evaluation principle is to holistically assess dramatic performance, rather than evaluating agent responses individually, turn by turn. This approach is crucial because drama often incorporates literary devices such as foreshadowing and plot twists. Consequently, a seemingly subpar or unusual generation in a single turn might be a deliberate narrative setup for subsequent developments and should not be penalized in isolation.

For the overall drama performance, we employ HAMLETJudge to automate the evaluation process. This model conducts pairwise comparisons between the test result and the baseline result, assigning a score based on a 5-point Likert scale (Robinson, 2014) to determine a win rate. The detailed scoring guideline is defined in Appendix B.

---

[1]https://m.douban.com/subject_collection/ECKM5FBEI
[2]https://sites.prh.com/modern-library-top-100/#top-100-novels

# 5 EXPERIMENTS

In this section, we present the experimental results to demonstrate the superiority of our proposed HAMLET framework, alongside ablation studies verifying its reliability and validity.

**Settings.** To ensure rigorous and reproducible evaluation, we define the baseline and test configurations clearly. All underlying models except the PAD component in HAMLET share the same GPT-4o backbone, with a greedy sampling strategy. Regarding user settings, each agent can be freely configured, and the PAD component is optional.

**HAMLET Leaderboard.** Following the settings above, we compared various mainstream LLMs ranging from open-source to closed-source and non-reasoning to reasoning models. Table 1 reveals their capabilities in both English and Chinese online drama performance, serving as a practical reference for real-world applications. Notably, Claude-4-sonnet-Thinking demonstrated exceptional proficiency across the majority of evaluated metrics in both languages, highlighting its versatility and effectiveness in dynamic, interactive theatrical environments.

**Scaling Laws and Reasoning Efficiency.** We observed that while scaling laws persist in drama scenarios, as evidenced by performance gains with model size (e.g., the Qwen3 series), reasoning capability proves to be more parameter-efficient than merely increasing model scale. For instance, Qwen3-32B-Thinking (73.85) significantly outperforms the much larger Llama-3.1-70B (45.75) and even rivals the massive Qwen3-235B (71.21). This suggests that in drama tasks, which demand multi-constraint satisfaction involving goals, persona, and context, enhanced reasoning density is more critical than simple parameter scaling.

**Subtext and Strategy.** Furthermore, reasoning models like DeepSeek-R1 and Claude-4-sonnet-Thinking consistently outperform their non-reasoning counterparts. We attribute this to their superior capacity for subtext processing and strategic planning. While standard models often react superficially to immediate dialogue, reasoning models utilize internal chain-of-thought to simulate future outcomes and deduce implied intent (e.g., "He is lying, but I should pretend to believe him to trap him later"). This cognitive depth significantly enhances character dimensionality and facilitates complex narrative devices such as plot twists and foreshadowing.

| Model | English | | | | Chinese | | | | Overall Score |
|---|---|---|---|---|---|---|---|---|---|
| | Character | Narrative | Interaction | Average | Character | Narrative | Interaction | Average | |
| *Non-Reasoning Models* | | | | | | | | | |
| Claude-4-sonnet | 76.50 | 77.30 | 76.96 | 76.92 | **80.18** | 79.20 | 79.66 | 79.68 | 78.30 |
| Claude-3.7-sonnet | 65.80 | 66.94 | 65.98 | 66.24 | 76.00 | 75.20 | 75.48 | 75.56 | 70.90 |
| Claude-3.5-sonnet | 61.00 | 62.15 | 62.10 | 61.75 | 60.50 | 59.00 | 59.99 | 59.83 | 60.79 |
| Gemini-2.5-flash | 46.00 | 47.10 | 46.70 | 46.60 | 52.00 | 52.90 | 52.15 | 52.35 | 49.48 |
| GPT-4.5-preview | 59.21 | 60.00 | 59.92 | 59.71 | 61.50 | 62.50 | 61.91 | 61.97 | 60.84 |
| GPT-4.1 | 63.50 | 62.49 | 62.98 | 62.99 | 62.00 | 62.80 | 62.64 | 62.48 | 62.74 |
| Mistral-Small-3.2-24B | 48.00 | 48.50 | 48.22 | 48.24 | 51.00 | 51.83 | 52.00 | 51.61 | 49.93 |
| DeepSeek-V3-0324 | 54.00 | 55.13 | 55.00 | 54.71 | 64.50 | 64.00 | 64.01 | 64.17 | 59.44 |
| Llama-3.1-70B | 49.80 | 49.00 | 48.86 | 49.22 | 42.00 | 42.81 | 42.00 | 42.27 | 45.75 |
| Llama-3.1-8B | 35.00 | 36.01 | 35.52 | 35.51 | 33.50 | 34.00 | 33.99 | 33.83 | 34.67 |
| Higgs-Llama-3-70B | 72.00 | **78.50** | 66.22 | 72.24 | 64.00 | 64.50 | 63.95 | 64.15 | 68.20 |
| Qwen3-8B | 47.50 | 46.80 | 47.00 | 47.10 | 58.00 | 57.50 | 58.05 | 57.85 | 52.48 |
| Qwen3-32B | 65.00 | 65.88 | 65.20 | 65.36 | 66.00 | 65.50 | 65.84 | 65.78 | 65.57 |
| Qwen3-235B-A22B | 69.50 | 69.80 | 69.65 | 69.65 | 72.50 | 73.00 | 72.78 | 72.76 | 71.21 |
| *Reasoning Models* | | | | | | | | | |
| Gemini-2.5-pro | 61.00 | 62.22 | 61.70 | 61.64 | 62.00 | 62.80 | 62.25 | 62.35 | 62.00 |
| Claude-4-sonnet-Thinking | **79.50** | 78.40 | **79.04** | **78.98** | 78.42 | **80.32** | **81.03** | **79.92** | **79.45** |
| Minimax-M1 | 51.50 | 52.32 | 52.00 | 51.94 | 65.00 | 65.50 | 65.19 | 65.23 | 58.59 |
| OpenAI-o3 | 69.00 | 69.95 | 69.40 | 69.45 | 78.00 | 77.50 | 78.17 | 77.89 | 73.67 |
| Magistral-Small-2506 | 59.00 | 60.00 | 59.74 | 59.58 | 58.50 | 59.30 | 58.90 | 58.90 | 59.24 |
| DeepSeek-R1-0528 | 66.00 | 67.10 | 66.64 | 66.58 | 79.00 | 79.50 | 79.61 | 79.37 | 72.98 |
| Qwen3-8B-Thinking | 50.00 | 51.61 | 51.00 | 50.87 | 65.80 | 65.00 | 65.55 | 65.45 | 58.16 |
| Qwen3-32B-Thinking | 69.50 | 68.80 | 69.00 | 69.10 | 78.00 | 79.00 | 78.77 | 78.59 | 73.85 |
| Qwen3-235B-A22B-Thinking | 70.50 | 71.00 | 70.72 | 70.74 | 76.00 | 75.80 | 75.96 | 75.92 | 73.33 |

Table 1: Performance evaluation of different models based on HAMLETJudge. **Bold** values indicate the best performance in each column.

| Metric | HamletJudge (ours) | GPT-4.1 | Claude-4-Sonnet | Gemini-2.5-Pro |
|---|---|---|---|---|
| CP | **0.792** | 0.675 | 0.698 | 0.720 |
| NQ | **0.807** | 0.593 | 0.783 | 0.684 |
| IE | 0.773 | 0.622 | **0.804** | 0.701 |
| Average | **0.791** | 0.630 | 0.762 | 0.702 |

Table 2: Pearson correlation coefficients (Pearson, 1901) of different models and human evaluators.

| Model (FC) | Responding Strategy | | | Latency Penalty | Final Score |
|---|---|---|---|---|---|
| | Fast | Slow | Silence | | |
| Qwen2.5-7B-Instruct | 0.779 | 0.359 | 0.131 | 0 | 0.423 |
| Qwen2.5-72B-Instruct | 0.692 | 0.452 | 0.262 | 0 | 0.469 |
| Qwen3-8B | 0.699 | 0.452 | 0.066 | 0 | 0.406 |
| Qwen3-32B | 0.773 | 0.396 | 0.098 | 0 | 0.422 |
| Qwen3-8B-Thinking | 0.668 | 0.321 | 0.459 | 0.05 | 0.532 |
| Qwen3-32B-Thinking | 0.668 | 0.434 | 0.361 | 0.10 | 0.538 |
| GPT-4o | 0.556 | 0.623 | 0.328 | 0.024 | 0.478 |
| GPT-4.1-mini | **0.909** | 0.132 | 0.180 | 0 | 0.407 |
| DeepSeek-R1-0528 | 0.723 | 0.615 | 0.470 | 0.15 | 0.453 |
| Gemini-2.5-pro | 0.742 | 0.536 | 0.519 | 0.10 | 0.449 |
| **PAD (ours)** | 0.822 | **0.736** | **0.711** | 0 | **0.756** |

Table 3: Evaluation under responding strategies by tool calling. Underlined values reflect an applied latency penalty.

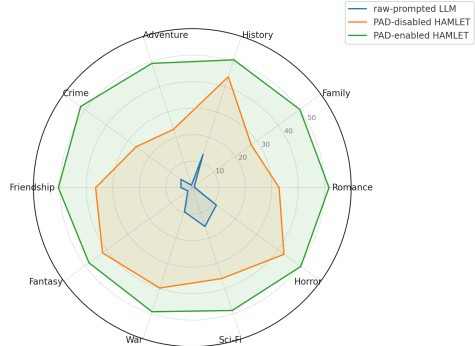

Figure 6: The ablation study results on PAD and multi-agent framework design of HAMLET. The radar chart illustrates the comparative evaluation of three experimental configurations in the context of online drama performance.

## 5.1 RELIABILITY OF HAMLET

Both HAMLETJudge and PAD were trained on data annotated by human labelers. To rigorously assess the reliability of our data, we conducted Inter-Annotator Agreement (IAA) analysis across annotators for both datasets. As demonstrated in Appendix G.3, the HAMLETJudge dataset achieves an overall weighted Krippendorff's Alpha of 0.725, while the PAD dataset yields an overall weighted Fleiss' Kappa of 0.624, both indicating substantial agreement. To further validate the effectiveness of HAMLETJudge, we measured its agreement with human ratings on a held-out validation set using the Pearson correlation coefficient. As reported in Table 2, HAMLETJudge closely aligns with human assessments and outperforms strong baselines such as GPT-4.1, Claude-4-sonnet, and Gemini-2.5-pro.

To demonstrate the reliability of PAD, we conducted additional experiments evaluating model performance under diverse response strategies. Notably, models with stronger general capabilities tend to exhibit more homogeneous and consistent performance across scenarios. However, models that generate reasoning tokens prior to tool calls often incur unacceptable latency in real-time drama settings. To account for this, we introduce a latency penalty that quantifies the impact of such delays on overall performance. Results are presented in Table 3.

Our analysis reveals a clear trade-off between performance and latency among existing models. Reasoning-intensive models achieve balanced performance across strategies but suffer from substantial latency penalties. In contrast, non-reasoning models are faster but display significant performance bias—performing well in fast-paced scenarios yet struggling in complex, nuanced interactions. PAD effectively reconciles this tension: it achieves the highest overall score while maintaining stable performance across all strategies, all with negligible latency. More discussion about the latency penalty can be found in Appendix G.4.

## 5.2 ABLATION STUDY

In the ablation study, we conducted experiments on three core component designs in HAMLET, to validate their individual contributions to overall system performance.

**Perceive and Decide Module.** We randomly select 30 topics from the evaluation dataset and control the experiment settings as GPT-4o under greedy sampling strategy, then compare three experimental setups: drama performed by (i) raw-prompted GPT-4o only; (ii) the multi-agent collaboration of the full HAMLET framework; and (iii) HAMLET framework without PAD.

As shown in Figure 6, prompting a single GPT-4o yields substantially lower performance, highlighting the necessity of our multi-agent workflow design. Furthermore, PAD-enabled HAMLET

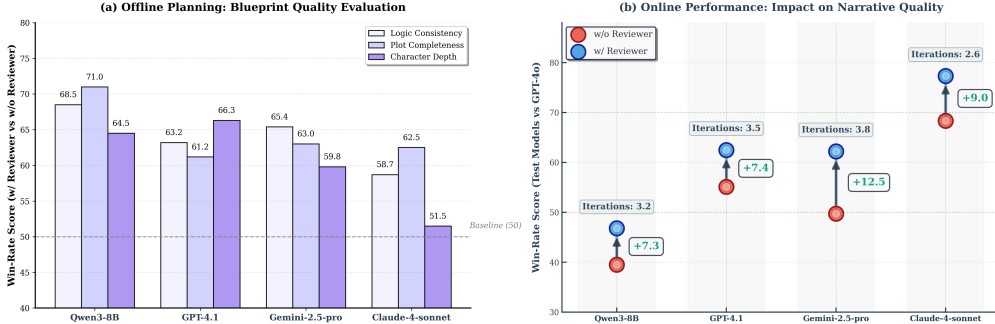

Figure 7: Ablation study of the reviewer across four test models. (a) Offline narrative blueprint quality is evaluated by human raters using win-rate. Scores above $50$ (dashed line) indicate a preference for blueprints generated with the reviewer. (b) Enabling the reviewer leads to a score raise of online Narrative Quality (from red to blue). Iterations denotes the average number of revision rounds performed with the reviewer.

consistently outperforms the version without PAD in all 10 topic categories, demonstrating that PAD serves as a crucial component in the online drama rendition, making the interaction and conversation of AI actors more natural, coherent and human-like.

**Reviewer.** To better assess the reviewer's contribution, we introduced dedicated evaluation metrics for the offline planning stage. As shown in Figure 7(a), the reviewer makes a significant contribution to improving the quality of narrative blueprints during offline planning. Intriguingly, we also observe in Figure 7(b) that the reviewer exerts a measurable influence on Narrative Quality during the online performance phase, enabling consistent substantial score improvements across all test models. We attribute this downstream effect to the causal linkage between blueprint quality and final narrative output: higher-fidelity planning directly enables more coherent, compelling, and structurally sound storytelling at execution time.

**Advancer.** Experiments show that removing the advancer reduces HAMLET 's task completion rate from 100% to 68.1%, highlighting its critical role in ensuring robustness and preventing conversational deadlocks. Additional experimental details and results are provided in Appendix I.

## 5.3 CASE STUDY

To further illustrate the operational mechanism of HAMLET and the role of each component, we present a set of real examples in Table 6 in Appendix C.

In Case 1, the narrator first assesses the situation before responding to make a reasonable judgment about the AI actor's input. HAMLET also supports users assuming specific character roles. In Cases 2, 3, and 4, the narrator, transfer, and advancer collaboratively handle challenging circumstances (such as missing props, irresponsible actions, or stubborn character choices) to ensure narrative coherence and smooth dramatic progression. Meanwhile, the planner is responsible for multi-trajectory design and review. As demonstrated in Cases 5 and 6, the same goal can be met by different beat trajectories as long as the process is reasonable.

## 6 CONCLUSION

In this paper, we introduced HAMLET, a multi-agent framework designed to address key challenges in AI-driven theatre. HAMLET generates a guiding narrative blueprint from a simple topic and provides agents with a PAD module, enabling autonomous thinking and physical environmental interaction. To assess performance, we established a comprehensive evaluation method and a leaderboard, where our specialized critic model, HAMLETJudge, achieved top-ranking results. Extensive experiments show that our approach creates expressive and coherent theatrical experiences, paving a new path toward autonomous and immersive interactive drama experiences.

## 7 ETHICS STATEMENT

This work relies on human annotation, which is critical for both the training and evaluation of our proposed models. The annotation of PAD and HAMLETJudge datasets was conducted by a team of five human experts, each with at least one year of professional experience in relevant fields such as creative writing or drama creation. To ensure high-quality annotations, we provided clear and task-specific instructions to all labelers. In addition, the datasets underwent rigorous safety checks to ensure no personally identifiable information or potentially problematic content is available. The labeling instructions can be found in Appendix E and F.

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

## A    DETAILED INFORMATION FOR EVALUATION DATASET

Detailed information of 100 cases selected for the evaluation dataset is listed in Table 4. The dataset comprises three parts: 1) excerpts from 25 Chinese literary works, 2) excerpts from 25 English literary works, and 3) 50 custom-authored drama topics. For the HAMLET Leaderboard evaluation result in Table 1, the score for both English and Chinese is calculated on a 75-item set, which consists of the 25 literary excerpts and 50 custom topics in the corresponding language.

## B    SCORING GUIDELINE FOR HAMLETJUDGE

We use HAMLETJudge for critic and scoring, as a judge model, it follows a 5-point Likert scale for pairwise comparisons. The specific definition for each score is detailed in Table 5.

| Score | Preference | Description |
|-------|-----------|-------------|
| 1 | Strong preference for Model A | Model A is significantly better. |
| 2 | Moderate preference for Model A | Model A is somewhat better. |
| 3 | Tie / No preference | Both models' output are roughly equivalent in quality. |
| 4 | Moderate preference for Model B | Model B is somewhat better. |
| 5 | Strong preference for Model B | Model B is significantly better. |

Table 5: 5-Point Likert scoring guideline used for HAMLETJudge. **Model A** refers to the baseline model, and **Model B** refers to the test model.

## C    CASE STUDY

Table 6 presents representative real interaction cases generated by HAMLET, showcasing the distinct roles of individual agents and their collaborative behavior during the whole process. Through these cases, we illustrate how HAMLET maintains narrative coherence, ensures logical consistency, and enables flexible dramatic progression in complex and unpredictable user interactions.

To begin with, Case 1 highlights Narrator's ability to interpret user intent under ambiguity. Specifically, it correctly associates the term "knife" with the existing prop "dagger," allowing the user's action to be executed successfully. This demonstrates HAMLET 's robustness in resolving lexical variations and grounding user input in the current scene context.

Building on this, Cases 2, 3 and 4 involve human user role-playing as AI actor, and reveal how the system handles a wide range of irregular or disruptive inputs. In Case 2, the Narrator identifies an inappropriate and non-existent prop ("rifle"), and rejects the action to preserve setting consistency. Similarly, Case 3 showcases how physically impossible actions, such as "flying," are filtered out based on the play's realistic constraints. Notably, Case 4 presents a more complex situation, where the user repeatedly insists on an invalid action. Here, HAMLET effectively coordinates among multiple agents: the Narrator detects the invalid target, the Transfer identifies unmet flags, and the Advancer eventually intervenes with adaptive guidance to resolve narrative deadlock. This case illustrates the system's ability to detect stagnation, maintain progression, and deliver context-sensitive instructions.

Additionally, Cases 5 and 6 demonstrate the crucial role of Planner, showcasing HAMLET 's support for multi-trajectory planning. Both cases revolve around solving a murder mystery, yet they employ different investigative strategies. In Case 5, the AI actor (Sherlock Holmes) uncovers clues through physical evidence at the crime scene. While in Case 6, the same conclusion is reached via testimonial interrogation of other characters. Despite following distinct paths, both trajectories lead to the same dramatic outcome and are validated by the Planner based on their internal logic and consistency. These cases underscore the system's flexibility in allowing diverse narrative flows, provided the progression remains coherent and goal-aligned.

| Selected established literature workpieces | |
|---|---|
| *1. Dream of the Red Chamber* | *2. Journey to the West* |
| *3. Romance of the Three Kingdoms* | *4. Water Margin* |
| *5. The Three-Body Problem* | *6. To Live* |
| *7. Four Generations Under One Roof* | *8. Memories of Peking: South Side Stories* |
| *9. Demi-Gods and Semi-Devils* | *10. The Smiling, Proud Wanderer* |
| *11. Wandering* | *12. Rickshaw Boy* |
| *13. Straw House* | *14. The Bronze Age* |
| *15. Border Town* | *16. The Chess Master* |
| *17. The Golden Age* | *18. My Father's Back* |
| *19. Blossoms* | *20. Frog* |
| *21. Cat Country* | *22. That Unknown Story* |
| *23. Farewell My Concubine* | *24. White Deer Plain* |
| *25. Fortress Besieged (Items 1–25 are translated titles of Chinese literary works.)* | *26. One Hundred Years of Solitude (Items 26–50 are original titles of English literary works.)* |
| *27. Brave New World* | *28. A Clockwork Orange* |
| *29. The Time Traveler's Wife* | *30. The Princess Bride* |
| *31. The Secret Garden* | *32. The Outsiders* |
| *33. The Call of the wild* | *34. Little Women* |
| *35. Hamlet* | *36. The Odyssey* |
| *37. Harry Potter* | *38. Frankenstein* |
| *39. The Kite Runner* | *40. King Lear* |
| *41. The tragedy of Macbeth* | *42. The Adventures of Huckleberry Finn* |
| *43. Life of Pi* | *44. A Tale of Two Cities* |
| *45. The tempest* | *46. Romeo and Juliet* |
| *47. The Adventures of Sherlock Holmes* | *48. Wuthering Heights* |
| *49. Catch-22* | *50. Don Quixote* |
| **Customizable drama topic design** | |

*51. Porco Rosso and Gina discuss topics about war, love and responsibility in a café, and after a while Phil also arrives.*

*52. Kenshin Himura, the wandering swordsman, walked into the café carrying his reverse-blade sword, only to find his late wife, Tomoe Yukishiro—who had died years ago saving him—standing there.*

*53. Conan and Gin engaged in a thrilling battle of deduction and a direct confrontation in the bustling Times Square, amidst the ebb and flow of countless passersby.*

*54. Furina and Herta met at the end of Sixth Avenue Alley, where they engaged in a profound debate about fate.*

*55. LeCun, Hinton, and Bengio engaged in an in-depth discussion during a NeurIPS coffee break about how AGI might be achieved and when it could arrive.*

*56. A wealthy man is murdered in his study, and the killer is among the guests present that night. Sherlock Holmes and Dr. Watson must unravel the mystery.*

*57. Lara Croft explores an ancient temple with Indiana Jones, debating the ethical implications of artifact removal.*

*58. Daenerys Targaryen and Jon Snow strategize their next move amidst the snowy battlements of Winterfell.*

*59. Tony Stark and Bruce Banner discuss the potential risks of AI development during a quiet night in the Avengers' tower.*

*60. Hermione Granger and Katniss Everdeen debate rebellion tactics in a secret library in a dystopian city.*

*61. Mario and Luigi race through a bustling New York subway station while evading Bowser's henchmen.*

*62. The Doctor from Doctor Who encounters Eleven from Stranger Things in a mysterious rift near Hawkins, Indiana.*

*63. Albert Einstein and Nikola Tesla debate the future of energy in a vintage café in Zurich.*

*64. Elsa from Frozen and Moana share stories of leadership and courage by the ocean shore during a summer festival.*

*65. Gandalf and Yoda discuss the nature of the Force and magic in a mystical forest clearing.*

*66. Nathan Drake and Sam Fisher team up to retrieve a stolen artifact in the crowded streets of Marrakech.*

*67. Elizabeth Bennet and Jay Gatsby engage in a witty conversation at a grand 1920s party.*

*68. Da Vinci and Michelangelo argue about art and innovation inside a Renaissance workshop.*

*69. Bruce Wayne and Clark Kent discuss justice and responsibility during a rainy night on a Gotham rooftop.*

*70. Katara and Zuko from Avatar: The Last Airbender reconcile old conflicts while watching a sunset by the river.*

*71. Mario and Princess Peach plan a secret mission to rescue Luigi from Bowser's castle under the moonlight.*

*72. Jon Snow and Arya Stark train together in the godswood of Winterfell, reflecting on their past journeys.*

*73. Neo and Trinity explore the Matrix's origins during a rare moment of calm in a futuristic cityscape.*

*74. Walter White and Jesse Pinkman discuss redemption and consequences in a dimly lit Albuquerque diner.*

*75. Daenerys Targaryen and Sansa Stark debate leadership styles during a council meeting in King's Landing.*

*76. Rick Grimes and Michonne survive and strategize while hiding in an abandoned shopping mall during a zombie apocalypse.*

*77. Loki and Thor bicker about family legacy while trapped in an ancient Norse temple.*

*78. Yennefer and Geralt of Rivia share a quiet moment at a bustling marketplace in Novigrad.*

*79. Miyamoto Musashi and Sun Tzu discuss the art of war on a foggy mountaintop.*

*80. Shrek and Donkey accidentally find themselves in a futuristic city, trying to find their way back to the swamp.*

*81. Katniss Everdeen and Peeta Mellark share a secret conversation in the Capitol's underground tunnels.*

*82. Sherlock Holmes and Irene Adler exchange clever banter at an exclusive London club.*

*83. Darth Vader and Luke Skywalker face off in a climactic duel inside the Death Star's throne room.*

*84. Elizabeth Bennet and Mr. Darcy meet unexpectedly at a winter ball in Regency England.*

*85. Professor McGonagall and Minerva McGonagall compare notes on magical education at Hogwarts.*

*86. Arthur Morgan and Dutch van der Linde plan their next heist while camping under the stars.*

*87. Geralt and Jaskier share songs and stories in a cozy tavern in the Northern Kingdoms.*

*88. Jon Snow and Tormund Giantsbane hunt in the frozen wilderness beyond the Wall.*

*89. Mario, Luigi, and Toad race through the Mushroom Kingdom to stop Bowser's latest scheme.*

*90. Tony Stark and Pepper Potts celebrate a rare peaceful evening at Stark Tower's rooftop garden.*

*91. Da Vinci and Galileo discuss the mysteries of the universe during a candlelit dinner.*

*92. Black Widow and Hawkeye reminisce about their past missions over coffee in a quiet New York café.*

*93. Frodo and Samwise rest beside the campfire, reflecting on their journey to Mount Doom.*

*94. Neo and Morpheus debate the ethics of free will inside the Matrix's control room.*

*95. Arya Stark and Gendry share a quiet moment forging weapons in Winterfell's smithy.*

*96. Link and Zelda strategize the defense of Hyrule Castle under threat from Ganondorf.*

*97. Mad Max and Furiosa race across the wasteland seeking a new safe haven.*

*98. Jesse Pinkman and Saul Goodman argue over legal and moral boundaries in a dingy Albuquerque office.*

*99. Bilbo Baggins hosts a surprise party in the Shire, attended by dwarves and elves alike.*

*100. Hannibal Lecter and Clarice Starling engage in a tense psychological game inside a mental institution.*

Table 4: The public dataset of established literary works and customized topic design list.

| Case | Description | Pieces of Real-time Drama Performance or Interaction results |
|------|-------------|-------------------------------------------------------------|
| 1 | AI actor Real-time Performance | *Case Abstract: AI actor with reasonable action input and existing props.*
Hamlet: (Grab a knife and step forward) You have no where to hide.
Narrator: `<think>`Detect two actions. Knife may refers to dagger, which exists in Hamlet's possession, first action success. Step forword is also reasonable under this condition, second action success.`</think>`**Hamlet paces agitatedly, dagger in hand.** |
| 2 | Human Player Roleplaying certain AI actor | *Case Abstract: Human player with non-existing props interaction.*
Hamlet: (Take out a riffle, aim at Claudis and pull the trigger) Say hello to my father, Claudius!
Narrator: `<think>`Riffle is a kind of gun, which is irrationality in current drama settings, and it does not exist in props list, action failure.`</think>`**Action failure, nothing happened.** |
| 3 | Human Player Roleplaying certain AI actor | *Case Abstract: Human player with irreasonable action input.*
Hamlet: (Take to the air and fly out of the palace) HAHAHA! I am superman.
Narrator: `<think>`Flying is physically impossible for humans without special equipment, and contradicts the realistic setting of the play. This action violates the established rules of the world, action failure.`</think>`**Action failure, nothing happened.** |
| 4 | Human Player Roleplaying certain AI actor | *Case Abstract: Human player with stubborned choice.*
Hamlet: (Take out dagger, stub Claudis).
Narrator: `<think>`Claudis does not exist in current scene`</think>`**Action failure, nothing happened.**
Environment: There are slight noises behind the curtain.
Hamlet: (Use dagger to stub Claudis).
Narrator: `<think>`Claudis does not exist in current scene, unlegal action.`</think>`**Action failure, nothing happened.**
Transfer: Polonius hasn't been stubbed, flag is not satisfied. (...repetitive patterns omitted...)
System: Time accumulation has surpassed the threshold, Advancer is activated.
Advancer: `<think>`I should give necessary instructions to make sure flag is met. The flag is ..., current situation is ..., current point is stucked because Hamlet's action.`</think>`Instruction to Hamlet: You should try to stab the curtain with your dagger.
Hamlet: (Stab the curtain)
Narrator: `<think>`Curtain exists in props, stab is a reasonable action. Action success`</think>`**Hamlet stabs through the curtain and pulls it back fiercely.** |
| 5 | AI actor Beat Trajectory - physical evidence | *Trajectory A: A wealthy man is murdered in his study, and the killer is among the guests present that night. (Searching the study for clues)*
Holmes: (Examining the desk.)
Narrator: Sherlock Holmes discovers a torn-up check.
Holmes: [This may hinting at a financial dispute](Inspecting the carpet.)
Narrator: Sherlock Holmes finds a cufflink that doesn't belong to the victim.
Holmes: (Checking the bookshelf)
Narrator: Sherlock Holmes notices a book slightly pulled out, revealing a hidden blackmail letter.
Holmes: Hmmm, I think I have the answer. The cufflink belongs to the butler. The blackmail letter proves the victim was threatening him. The check confirms the butler was desperate for money. The butler had motive, opportunity, and a direct link to the crime scene.
Planner: `<think>`(detailed analysis...)To conclude, the evidence acquisition process is complete and the reasoning is sufficient.`</think>`Trajectory check passed. |
| 6 | AI actor Beat Trajectory - testimony evidence | *Trajectory B: A wealthy man is murdered in his study, and the killer is among the guests present that night. (Conversing with key guests)*
Holmes: (Approaching the butler, eyes fixed on his sleeves) Where were you between nine and midnight?
Butler: "I—I was inventorying wine in the kitchen, sir."
Holmes: [His left cufflink is missing... precisely like the one found near the body.]
Holmes: (Turning abruptly to the maid) Did your employer argue with anyone recently?
Maid: (Wringing her apron) "He and Mr. Bates—the butler—had shouting matches all week... something about money..."
Holmes: [Financial disputes align with the torn check.]
Holmes: (Cornering the victim's business partner near the fireplace) Did the deceased mention threats?
Business Partner: (Forced laugh) "Oh, he loved boasting—said he 'had Bates by the throat' last Tuesday."
Holmes: [A clear admission of blackmail.]
Holmes: (Striding to the center of the room) The butler's missing cufflink places him at the crime scene. The maid confirms violent quarrels over money, while our 'helpful' friend here reveals the victim's blackmail scheme. Motive, opportunity, and physical evidence—all point to one man.
Planner: `<think>`(detailed analysis...)To conclude, the evidence acquisition process is complete and the reasoning is sufficient.`</think>`Trajectory check passed. |

Table 6: Case study of HAMLET that illustrates the roles and collaborative processes of each agent during online performance.

Taken together, these case studies demonstrate the effectiveness of HAMLET in managing interactive drama through robust input interpretation, constraint enforcement, multi-agent collaboration, and trajectory-level planning.

## D    DETAILED EXPLANATION OF THE OPERATION PROCESS OF PAD

This section provides a more detailed breakdown of the operational workflow for PAD. The entire process is formally presented in Algorithm 2.

PAD operates by first perceiving the complete dramatic context. This includes the current scenario $\mathcal{S}$, the actor profiles $\mathbb{C}$ and typically goals of all actors $\mathbb{G}_c$ defined in profile $\mathbb{C}$, the set of interactable objects $\mathbb{O}$, and dialogue history $\mathcal{H}_d$. Based on this comprehensive input, PAD then decides the most

appropriate tool calls for each relevant actor. This decision manifests as two possible outputs: a speech pace $r$ and a potential physical action $o$, formulated as a Subject-Verb-Object triplet. This perceive-decide cycle repeats, enabling dynamic and context-aware character behaviors.

# E    LABELING INSTRUCTION FOR HAMLETJUDGE

## E.1    TASK OVERVIEW

Your task is to compare two complete drama generations from two different Large Language models (referred to as Model A and Model B). For each task, you will be presented with:

1. Two complete drama performance results generated by **Model A** and **Model B**.
2. Current target **Evaluation Dimension** and its **Corresponding Criteria**.

Your core job is to determine which model performed better according to the given dimension and criteria, provide a detailed justification for your choice, and assign a score based on a 5-point comparative scale.

## E.2    EVALUATION DIMENSIONS

You will be asked to judge the models on one of three key dimensions for each task.

### CHARACTER PERFORMANCE (CP)

This dimension assesses the depth, consistency, and believability of the characters. When evaluating CP, consider:

- **Consistency:** Are the characters' actions and dialogues consistent with their established personalities, motivations, and backgrounds? Do they act "out-of-character"?
- **Depth & Development:** Are the characters multi-dimensional and evolving, or are they flat, one-note stereotypes? Do they show growth or reveal new aspects of their personality during the performance?
- **Believability:** Does the dialogue sound natural for the characters? Are their emotional reactions and decisions plausible within the context of the scene?

### NARRATIVE QUALITY (NQ)

This dimension focuses on the story's structure, creativity, and engagement. When evaluating NQ, consider:

- **Plot Advancement:** Does the story move forward effectively, or does it stall with filler content and irrelevant actions? Does it build towards a meaningful conclusion?
- **Creativity & Engagement:** Is the narrative novel, clever, and surprising? Does it spark curiosity and make you want to know what happens next, or is it predictable and dull?
- **Coherence & Logic:** Is the plot internally consistent? Are plot twists and developments well-founded, or do they feel random and nonsensical, breaking the narrative's logic?

### INTERACTION EXPERIENCE (IE)

This dimension evaluates the overall flow, pacing, and immersive quality of the drama—essentially, its "readability" and "feel." When evaluating IE, consider:

- **Flow & Pacing:** Are the transitions between scenes and character interactions smooth? Is the pacing effective—building tension and providing moments of reflection appropriately—or does it feel rushed or dragged out?
- **Content Effectiveness:** How much of the text is meaningful versus repetitive or vague filler? Which response is more concise and impactful in its delivery?

---

**Algorithm 2** Detailed Working Algorithm for Perceive and Decide Module

---

1: **procedure** PADMODULE
2:     **Drama Context Configuration:**
3:         $\mathcal{T}$: *Drama Topic*
4:         $\mathcal{A}_i$: *Current Act*
5:         $\mathcal{P}_j$: *Current Point*
6:
7:     **For $\mathcal{P}_j$ exists:**
8:         $\mathcal{S}$: *Current Scenario*
9:         $\mathbb{A} = \{a_1, ..., a_n\}$: *Actor List*
10:         $\mathbb{C} = \{c_1, ..., c_n\}$: *Actor Profile*
11:         $\mathbb{G}_c = \{g_1, ..., g_n\}$: *Actor Goal*
12:         $\mathbb{O} = \{o_1, ..., o_m\}$: *Interactable Objects*
13:         $\mathcal{H}_d$: *Dialogue History*
14:         $\pi_\theta$: *Model Parameters and Sampling Strategy*
15:
16:     **Output:**
17:         $r \in \{\texttt{FAST}, \texttt{SLOW}, \texttt{SILENCE}\}$
18:         $o \in \{\varnothing, (\texttt{s}, \texttt{v}, \texttt{o})\}$
19:
20:     **Initialize:**
21:         $t_0 \leftarrow CurrentTime()$
22:         $\mathcal{H}_{d_0} \leftarrow GetHistory(t_0)$
23:         $\delta = \texttt{IsActCompleted}(\mathcal{A}_i) \leftarrow \texttt{False}$
24:
25:     **while** $\delta = False$ **and** $\mathcal{H}_{d_t} \neq \mathcal{H}_{d_0}$ **do**
26:         $H_{new} \leftarrow GetHistory(t - t_0)$
27:         $\mathcal{H}_{d_t} \leftarrow \mathcal{H}_{d_t} \cup \{H_{new}\}$
28:         **for all** $a_k \in \mathbb{A} \cap \{a_{\text{current}}\}^c$ **do**
29:             $prompt \leftarrow \texttt{Encode}(\mathbb{C}, \mathbb{G}_c, \mathbb{O}, \mathcal{S}, \mathcal{H}_{d_t})$
30:             $(\hat{r}_{a_k}, \hat{o}_{a_k}) \leftarrow \text{PAD}(prompt, \pi_\theta)$
31:             $r_{a_k} \leftarrow \texttt{ParseResponse}(\hat{r}_{a_k})$
32:             $o_{a_k} \leftarrow \texttt{ParseAction}(\hat{o}_{a_k})$
33:         **end for**
34:     **end while**
35:     $t \leftarrow t + \Delta t$
36:     **return** $\{(r_{a_k}, o_{a_k}) : a_k \in \mathbb{A}\}$
37: **end procedure**
38:
39: **Helper Functions:**
40: **function** PARSERESPONSE($\hat{r}$)
41:     **return** $\arg\max_{r \in \{\texttt{FAST}, \texttt{SLOW}, \texttt{SILENCE}\}} P(r|\hat{r})$
42: **end function**
43: **function** PARSEACTION($\hat{o}, a_k$)
44:     **if** $\hat{o}$ indicates no action **then**
45:         **return** $\varnothing$
46:     **else**
47:         $s \leftarrow a_k$
48:         $v \leftarrow \texttt{ExtractVerb}(\hat{o})$
49:         $o \leftarrow \texttt{ExtractObject}(\hat{o})$
50:         **if** $v = \varnothing$ **or** $o = \varnothing$ **then**
51:             **return** $\varnothing$
52:         **end if**
53:         **if** $o \in \mathbb{O}$ **then**
54:             **return** $(s, v, o)$
55:         **else**
56:             **return** $\varnothing$
57:         **end if**
58:     **end if**
59: **end function**

- **Overall Immersion:** Which version feels more like a genuine piece of theatre? Which one is more successful at making you forget you are reading an AI-generated script and helps you become absorbed in the story?

### E.3 RATING SCALE AND OUTPUT FORMAT

Please use the 5-point Likert scale for your pairwise comparison and follow the output format strictly.

### SCORING GUIDELINES

- **1: Strong preference for Model A** - Model A is significantly better than Model B.
- **2: Moderate preference for Model A** - Model A is somewhat better than Model B.
- **3: Tie** - Both models are of roughly equal quality, or their strengths and weaknesses balance each other out.
- **4: Moderate preference for Model B** - Model B is somewhat better than Model A.
- **5: Strong preference for Model B** - Model B is significantly better than Model A.

### OUTPUT FORMAT

Your output must be in the following format:

```
explanation: <A detailed explanation of your choice.
You must reference the specific evaluation dimension
and provide concrete examples or quotes from both
models' outputs to justify your reasoning. Directly
compare the strengths and weaknesses that led to your
score.> score: <1, 2, 3, 4, or 5> choice: <Model A,
Model B, or tie>
```

### E.4 EXAMPLE

Here is a complete annotation example to illustrate the task.

---

**Current Evaluation Dimension:** Narrative Quality (NQ)

**Drama Topic:** A detective investigates a stolen artifact in a museum after hours.

**Model A Result:**

Detective Smith entered the dark exhibition hall, turning on his flashlight. The beam cut across an empty pedestal. "It was stolen from here," he muttered. On the floor, he saw a faint footprint and knelt to inspect it. "About a size 9," he noted. He then went to question the night guard, who claimed he didn't hear a thing all night.

**Model B Result:**

Upon entering the hall, Detective Smith's first glance wasn't at the empty pedestal, but up at the ceiling's security camera. "Odd," he murmured, "the camera was deactivated last night, but the log shows an internal authorization." Tucked beneath the pedestal, he found a small, folded note with a riddle: "I sing where there is light, but die where there is silence."

**Your Output:**

```
explanation:  I have a strong preference for Model
B. The evaluation dimension is Narrative Quality.
Model A's plot is highly generic and follows a clichéd
procedural path (inspecting a footprint, questioning
the guard) that fails to build any real suspense.
In contrast, Model B immediately establishes a more
complex and engaging mystery by introducing the
"internally authorized" deactivation of the camera,
hinting at an inside job.  Furthermore, the inclusion
of the cryptic riddle is an excellent narrative hook
that is far more creative and provides an imaginative
direction for the investigation.  Therefore, Model B
is significantly better in terms of Narrative Quality.

score:  5
choice:  Model B
```

### E.5 IMPORTANT CONSIDERATIONS

- **Focus on Substance, Not Nitpicks:** Please ignore minor grammatical errors or slightly awkward phrasing if the core intent and narrative substance of the response are clear.
- **No Single Correct Path:** Stories can evolve in many valid ways. The goal is not to enforce a single "correct" storyline, but to reward the model that tells a *more compelling, creative, and well-executed* story.
- **Trust Your Judgment:** The line between "somewhat better" and "significantly better" can be subjective. Use your best judgment based on the criteria above and strive to be consistent in your evaluations.
- **Quality Assurance:** If you are uncertain about the annotation result, please leave it blank rather than making an uncertain guess.

## F ANNOTATION GUIDELINES FOR PAD

### F.1 TASK OVERVIEW

PAD determines the underlying strategy of a response for each AI actor. It is inspired by human cognition, choosing between a fast, intuitive reaction (System I) and a slow, deliberate one (System II), with the additional option of strategic silence and potential actions.

For each task, you will be presented with a complete snapshot of a character's decision-making moment, structured to reflect both internal state and external stimulus as defined in PAD:

- **Internal State** — The character's self-aware profile, comprising:
  - *Persona*: Core personality traits, background, and identity.
  - *Subjective Relationships*: The character's personal perceptions of and emotional ties to other actors.
  - *Goal*: The character's current objective or intention.
  - *Memory*: Summarized recollections of past events and interactions, distilled from prior scenes and retrieved via Retrieval-Augmented Generation (RAG).
- **External Stimulus** — The objective contextual information available in the scene, including:
  - *Environment Description*: A depiction of the physical or situational setting (e.g., a dimly lit alley, a tense courtroom).
  - *Actor List*: A real-time roster of all characters present, including their roles and observable states (note: this reflects external perception only, distinct from the subjective relationships in the internal state).

- – *Dialogue History*: The chronological record of all prior spoken or written exchanges, including tone, unresolved tensions, commitments, and relevant subtext.
  - – *Interactable Objects*: Physical or digital items in the environment that the character may use, reference, or react to—each potentially imbued with functional, symbolic, or emotional significance (e.g., a flickering lantern, a locked diary, a ringing phone).
- **Final Response** — The most appropriate responding strategy in tool calling format, grounded in the integration of internal state and external stimulus.

## F.2    RESPONDING STRATEGIES: DESCRIPTION AND DEFINITION

The allowed tool calling format for PAD comprises two main categories: Single tool use, which includes *Fast*, *Slow*, and *Silence*, and Multi tool use, which involves combinations of *Action + Fast*, *Action + Slow*, and *Action + Silence*.

### F.2.1    FAST (INTUITIVE REACTION)

A fast response is driven by instinct, emotion, and immediate impressions.

- **Characteristics:** The decision is impulsive, emotional, or based on a gut feeling. The reasoning is often simple, relying on stereotypes, recent events, or strong emotional states (e.g., anger, surprise, excitement).
- **When to Choose This:** Select this option if the character's *Thinking* and final *Response* reflect an immediate, unfiltered reaction that bypasses deep strategic analysis. It should feel like a knee-jerk response, whether clever or foolish.

### F.2.2    SLOW (CONSIDERED DELIBERATION)

A slow response is analytical, rational, and goal-oriented.

- **Characteristics:** The decision is the result of weighing pros and cons, considering long-term consequences, recalling distant memories, or formulating a multi-step plan. The reasoning is complex and logical.
- **When to Choose This:** Select this option if the *Thinking* explicitly shows a process of deliberation. The character is clearly analyzing the situation, managing their impulses, and acting based on a calculated strategy to achieve a specific goal.

### F.2.3    SILENCE (STRATEGIC NON-RESPONSE)

Silence is not merely the absence of a response; it is a deliberate, tactical choice.

- **Characteristics:** The decision to say nothing is used to achieve a specific purpose, such as showing disapproval, building suspense, asserting dominance, avoiding a trap, or making another character uncomfortable.
- **When to Choose This:** Select this option when the most powerful or intelligent move in the given context is to not respond. The *Thinking* process should ideally reveal a conscious, strategic reason for choosing silence over speech.

### F.2.4    ACTION

Action represents a non-verbal, physical, or environmental intervention performed by the character—such as picking up an object, moving toward another actor, gesturing, or manipulating the scene. Unlike dialogue-based responses, actions convey intent through behavior and can carry significant narrative or strategic weight.

- **Characteristics:** Actions may be subtle (e.g., glancing at a door, clenching a fist) or overt (e.g., slamming a book, handing someone a key). They often serve to reinforce, contradict, or replace verbal communication, and should be grounded in the character's internal state (e.g., goal, emotion) and the external stimulus (e.g., interactable objects, social dynamics).

- **When to Choose This:** An *Action* should be annotated in combination with one of the three core responding strategies (*Fast*, *Slow*, or *Silence*) whenever the character's response involves a meaningful behavioral component beyond speech. The combination reflects how cognition and behavior jointly shape the character's agency:
    - **Action + Fast**: The character executes an impulsive physical reaction (e.g., flinching at a loud noise, grabbing a weapon in panic).
    - **Action + Slow**: The character performs a deliberate, premeditated act as part of a calculated plan (e.g., quietly pocketing a clue while distracting others).
    - **Action + Silence**: The character uses non-verbal behavior as a substitute for speech to exert influence or control (e.g., turning their back to signal rejection, slowly pouring a drink to delay answering).
- **Annotation Rule:** Always pair *Action* with exactly one of {*Fast*, *Slow*, *Silence*}. Do not annotate *Action* alone.

## F.3 ANNOTATION PROCESS AND EXAMPLE

You are presented with a complete decision-making snapshot for a focal character, as defined in Appendix F.1. Your task is to select the appropriate responding strategy (or strategy combination) that best characterizes the cognitive and behavioral mode underlying the character's response.

The annotation proceeds in three steps:

1. **Interpret the context:** Review the internal state (Persona, Subjective Relationships, Goal, Memory) and external stimulus (Environment Description, Actor List, Dialogue History, Interactable Objects) to understand the character's motivations, constraints, and situational opportunities.
2. **Embody the character:** Put yourself in the character's position. Consider how they would genuinely perceive, feel, and react in this moment, given who they are and what is happening around them. This empathetic reasoning should guide your judgment of their likely response mode—even if you do not explicitly write out their internal monologue.
3. **Generate the *Final Response* and label the strategy:** Produce the character's concrete output—dialogue, action, or silence—and tag it using one of the six allowed tool-calling formats: *Fast*, *Slow*, *Silence*, *Action + Fast*, *Action + Slow*, or *Action + Silence*.

**Example.** Consider a scene in a dimly lit study during a family inheritance dispute. The focal character, *Eleanor*, has just been accused of hiding a will by her brother, *Marcus*.

- **Internal State:**
    - *Persona:* Principled, introverted, values honesty but avoids confrontation.
    - *Subjective Relationships:* Resents Marcus for his domineering attitude; trusts her sister-in-law, Clara.
    - *Goal:* To clear her name without escalating the conflict.
    - *Memory:* Recalls Marcus falsely accusing her of stealing jewelry years ago.
- **External Stimulus:**
    - *Environment Description:* A cluttered Victorian study; rain taps against the windows.
    - *Actor List:* Marcus (aggressive, standing), Clara (anxious, seated), Lawyer (neutral, observing).
    - *Dialogue History:* Marcus: "You're the only one who had access—hand over the will, Eleanor."
    - *Interactable Objects:* A locked desk drawer, a half-empty teacup, a family photo on the mantel.

**Thinking (for illustration only):** *He's doing it again—twisting the truth to make me look guilty. I could shout back, but that's what he wants. If I stay calm, Clara will see through him. The will is in the drawer, but I won't give him the satisfaction of fetching it while he's yelling. Let him sweat for a moment. I'll wait, then offer to show everyone together—fair and quiet.*

**Final Response:** [Silence. Eleanor slowly picks up the teacup, takes a measured sip, and places it back on the saucer without looking at Marcus.]

**Strategy:** *Action + Silence*

**Rationale:** This response uses a controlled physical gesture (sipping tea) in conjunction with deliberate silence to project composure and assert quiet resistance. Although the underlying reasoning is strategic and reflective (akin to *Slow* cognition), the observable behavior prioritizes non-engagement—making *Action + Silence* the appropriate label, as the silence itself serves as the primary tactical choice.

### F.4 IMPORTANT CONSIDERATIONS

- **Strategy Over Style:** The literary quality or surface fluency of the *Final Response* should not influence your strategy selection. Focus on the *cognitive and behavioral mode* that best explains how the character arrived at their response. For instance, a clever one-liner may still be a *Fast* response if it stems from impulsive emotion rather than deliberate planning—even if the character is generally thoughtful. Conversely, a mundane reply can be *Slow* if it reflects calculated restraint.

- **Dramatic Plausibility:** The chosen strategy should be both *psychologically grounded* in the character's internal state and *narratively compelling*. Prioritize responses that feel authentic to the character's persona, relationships, and goals, while also advancing tension, subtext, or emotional stakes. A bold, surprising choice is preferable to a generic one—so long as it remains consistent with who the character is.

- **Anchor in Persona and Goal:** When multiple strategies seem plausible, resolve ambiguity by prioritizing the character's core `PERSONA` and current `GOAL`. Ask: "Given who this character is and what they want right now, which mode of response is most likely?" This principle overrides minor inconsistencies in dialogue tone or situational pressure.

- **Avoid Speculative Annotations:** If the context is insufficient to confidently determine a strategy—or if the character's internal state is too ambiguous—do not guess. Leave the strategy field blank and flag the instance for review. Erring on the side of caution preserves data quality and supports reliable model training.

## G TRAINING AND EVALUATION DETAILS FOR PAD AND HAMLETJUDGE

### G.1 TRAINING SETTINGS AND HYPERPARAMETERS

Both PAD and HAMLETJudge were fine-tuned based on Qwen3-8B using the LLaMA-Factory framework. Training was conducted with ZeRO-3 across $8 \times$ NVIDIA A800-SXM4-80GB GPUs, utilizing bfloat16 precision. The maximum sequence length was set to 65,536, and the total batch size was 128.

For PAD, we performed supervised fine-tuning (SFT) on the training set for 3 epochs. The optimizer was configured with a learning rate of 1e-5, a cosine learning rate scheduler, and a warmup ratio of 0.1.

For HAMLETJudge, we adopted a two-stage training paradigm. In the first stage, we conducted supervised fine-tuning on the first half of the training set, using the same optimization setup as PAD (learning rate of 1e-5, cosine scheduler, and 0.1 warmup ratio). In the second stage, leveraging the pairwise nature of the dataset, we restructured the remaining half of the training set into chosen–rejected format and applied Direct Preference Optimization (DPO). The DPO stage was trained for 2 epochs with a learning rate of 5e-6, a KL regularization coefficient $\beta = 0.1$, and the same cosine learning rate scheduler.

### G.2 DATA STATISTICS AND HOLDOUT STRATEGY

The complete PAD dataset comprises two main categories: Single Tool Use, which includes *Fast*, *Slow*, and *Silence* responses, and Multi Tool Use, which involves combinations such as *Action + Fast*, *Action + Slow*, and *Action + Silence*. To ensure data quality, we retained only instances

where at least four out of five expert annotators agreed on the strategy label; all other instances were discarded. Detailed statistics for each retained category are presented in Table 7.

| PAD | Single Tool Use | | | Multi Tool Use | | |
|---|---|---|---|---|---|---|
| | **Fast** | **Slow** | **Silence** | **Action+Fast** | **Action+Slow** | **Action+Silence** |
| Data Count | 4200 | 3304 | 3908 | 1000 | 1000 | 1000 |
| Mapping Rule | speech | **[thinking]**+speech | no speech | (*action*)+speech | (*action*)+ **[thinking]**+speech | (*action*) |

Table 7: Statistics of the PAD Dataset.

The **Mapping Rule** row in Table 7 illustrates the expected response formats for AI actors under each tool-use strategy. The core objective of PAD is to train agents in selecting appropriate response strategies, namely Fast, Slow, or Silence. In multi-tool use settings, the Action Tool is further annotated with a verb-object pair to reflect its parametrized usage in combination with a primary response strategy.

Importantly, during evaluation experiment as illustrated in Table 3, we exclude metrics associated with the Action Tool for the following reasons: 1) The Action Tool exhibits a high degree of freedom — whether or not to use it can both be justifiable under identical scenarios. 2) The choice of parameters (i.e., verb and object) is flexible and often has multiple valid options.

Given these characteristics, we consider the Action Tool as an auxiliary enhancement to the core response strategy in PAD, rather than a primary focus of quantitative evaluation. The dataset was partitioned via the holdout method (95% training, 5% testing), with stratification to preserve class distribution.

On the other hand, the HAMLETJudge dataset is designed to evaluate drama performance across Character Performance, Narrative Quality, and Interaction Experience. For each LLM pair (e.g., Claude-4-sonnet vs. GPT-4o), 75 pairwise comparisons are conducted per dimension in both Chinese and English, resulting in 150 annotated instances per dimension for each model pair.

We select 7 representative LLMs exhibiting diverse levels of drama performance, as demonstrated on the HAMLET Leaderboard: Claude-4-sonnet-Thinking, DeepSeek-R1-0528, Qwen3-235B-A22B, Gemini-2.5-pro, GPT-4o, Qwen3-8B, and LLaMA-3.1-8B. This leads to a total of $\binom{7}{2} = 21$ unique model pairs. With 150 data points per pair and 3 evaluation dimensions, the final dataset comprises $21 \times 150 \times 3 = 9{,}450$ annotated instances.

Detailed statistics are presented in Table 8.

| **HAMLETJudge** | **#CP** | **#NQ** | **#IE** | **Total** |
|---|---|---|---|---|
| Instance Number | 3150 | 3150 | 3150 | 9450 |

Table 8: Statistics of the HAMLETJudge Dataset.

To ensure fair evaluation, the dataset is splitted using a holdout strategy, with around 95.3% allocated for training and 4.7% for testing. Stratified sampling is applied to maintain a balanced distribution across all three dimensions.

### G.3 INTER-ANNOTATOR AGREEMENT ANALYSIS OF DATA

To quantify the reliability of both dataset, we conducted an Inter-Annotator Agreement (IAA) analysis using statistical measures tailored to the nature of each dataset. For the HAMLETJudge dataset, annotators assigned ratings on Likert-type scales for each dimension—ordinal data for which Krippendorff's Alpha ($\alpha$) is the appropriate reliability metric. As shown in Table 9, the overall weighted $\alpha$ is 0.725, indicating substantial agreement. Even on the most subjective dimension, Interaction Experience (IE), agreement remains high ($\alpha = 0.703$).

| Category (Dimension) | Data Count | Krippendorff's Alpha ($\alpha$) |
|---|---|---|
| Character Performance (CP) | 3,150 | 0.7485 |
| Narrative Quality (NQ) | 3,150 | 0.7241 |
| Interaction Experience (IE) | 3,150 | 0.7030 |
| Overall Weighted $\alpha$ | 9,450 | 0.7252 |

Table 9: Inter-Annotator Agreement (Krippendorff's Alpha) for the HAMLETJudge dataset.

| Strategy | Data Count | Fleiss' Kappa ($\kappa$) |
|---|---|---|
| Fast | 4,200 | 0.6612 |
| Slow | 3,304 | 0.6310 |
| Silence | 3,908 | 0.5780 |
| Overall Weighted $\kappa$ | 14,412 | 0.6240 |

Table 10: Inter-Annotator Agreement (Fleiss' Kappa) for the PAD dataset on Single Tool Use strategies.

For the PAD dataset, annotations consist of nominal, mutually exclusive strategy labels, making Fleiss' Kappa ($\kappa$) the suitable measure of agreement. As shown in Table 10, the dataset achieves an overall weighted $\kappa$ of 0.624, reflecting strong consensus among annotators. Together, these results suggest that, despite the inherent subjectivity of dramatic quality, our annotation guidelines successfully standardized expert judgments across both evaluation paradigms.

### G.4  PAD LATENCY PENALTY

As discussed in the PAD reliability experiments, although models that output reasoning tokens before invoking tool calls generally achieve better performance, this strategy may introduce intolerable extra latency in real-time drama settings. To address this trade-off, we introduce a latency penalty to balance PAD evaluation results.

Accurately and fairly measuring latency is inherently challenging due to several factors: 1) Different model sizes demand varying CUDA memory usage; 2) Certain models, such as DeepSeek-V3 and DeepSeek-R1, are recommended to use FP8 for inference, while most others typically rely on BF16; 3) For closed-source models, latency must be measured via API-level request-response timing, which is subject to various uncontrollable factors.

Despite these limitations, we still observe statistically significant latency differences across models. Therefore, to ensure consistency, we conduct all open-source model latency evaluations using 8 $\times$ NVIDIA H200 Tensor Core GPUs (just for sufficient CUDA memory) and deploy models with the vLLM framework, using default settings unless officially recommended sampling strategies are provided.

To quantify the impact of delay on user experience, we define a **Logarithmic Continuous Latency Penalty**. This metric is grounded in HCI perceptual thresholds. Unlike rigid step functions, our approach utilizes logarithmic transition segments to avoid artificial boundaries, reflecting the gradual nature of human perception. The refined penalty function $P(t)$ is defined as:

$$
P(t) = \begin{cases}
0 & t < 1.5s \\
0.05 \cdot \frac{\ln(1+t-1.5)}{\ln(2)} & 1.5s \leq t < 2.5s \\
0.05 & 2.5s \leq t < 8.0s \\
0.05 + 0.05 \cdot \frac{\ln(1+t-8.0)}{\ln(5)} & 8.0s \leq t < 12.0s \\
0.10 & 12.0s \leq t < 25.0s \\
0.10 + 0.05 \cdot \frac{\ln(1+t-25.0)}{\ln(11)} & 25.0s \leq t < 35.0s \\
0.15 & t \geq 35.0s
\end{cases} \tag{1}
$$

This formulation ensures that the penalty grows rapidly at the beginning of transition intervals but gradually saturates, effectively modeling the diminishing perceptual impact of additional delay. The visualization of this function is presented in Figure 8.

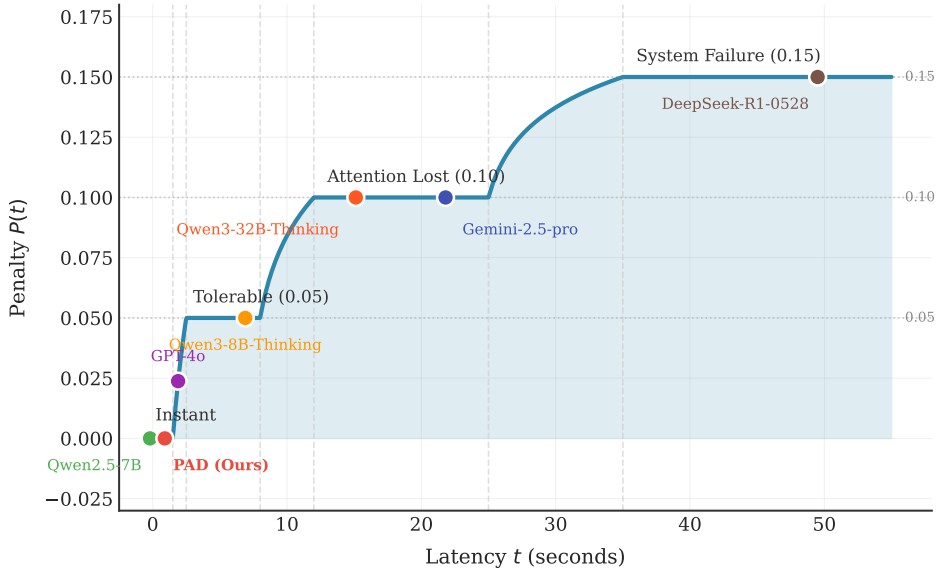

Figure 8: Visualization of Logarithmic Continuous Latency Penalty Function. The curve maps response latency to a penalty score based on HCI cognitive thresholds: Instant (penalty=0), Tolerable (0.05), Attention Lost (0.10), and System Failure (0.15). Logarithmic transitions are used between stages to model the gradual nature of user perception. Key models are annotated, showing that PAD (red dot) and small-size LLMs like Qwen2.5-7B-Instruct remain in the "Instant" zone, whereas reasoning-intensive models (e.g., DeepSeek-R1) incur higher penalties due to increased latency.

We apply this metric to our experimental data. The average latency and corresponding logarithmic penalty for each model are presented in Table 11. Notably, ultra-fast models like PAD and Qwen2.5-7B-Instruct incur zero penalty, whereas reasoning-intensive models receive a graded penalty commensurate with their processing delay.

| Model Name | Average Latency (s) | Log Penalty |
|---|---|---|
| Qwen2.5-7B-Instruct | 0.32 | 0 |
| **PAD (Ours)** | 0.36 | 0 |
| Qwen3-32B | 0.41 | 0 |
| GPT-4.1-mini | 0.75 | 0 |
| Qwen2.5-72B-Instruct | 1.02 | 0 |
| GPT-4o | 1.89 | 0.024 |
| Qwen3-8B-Thinking | 6.89 | 0.05 |
| Qwen3-32B-Thinking | 15.12 | 0.10 |
| Gemini-2.5-pro | 21.80 | 0.10 |
| DeepSeek-R1-0528 | 49.50 | 0.15 |

Table 11: Experiment results of average model latency and corresponding logarithmic penalty. The penalty is calculated using Equation 1, ensuring a smooth evaluation of latency costs in real-time drama settings.

## H  REAL-TIME FEASIBILITY AND COMPUTATIONAL COST

HAMLET is designed for practical, live deployment. To ensure real-time responsiveness, we implemented several engineering optimizations:

**Parallel Execution.** The multi-agent architecture supports asynchronous inference, allowing the Narrator and Planner to process context while Actors are generating responses.

**Efficient Inference.** We leverage the vLLM framework to maximize throughput. Furthermore, we employed INT4 quantization for the PAD model on NVIDIA H20 GPUs. Our validation demonstrates that this significantly reduces inference time and memory overhead while maintaining high decision quality, with only a 1.4% performance drop on the benchmark compared to the original FP16 model.

**PAD Efficiency.** PAD, a 8B model that we proposed, eliminates the latency overhead seen in reasoning models while maintaining high decision quality (Table 3), making it specifically optimized for low-latency interactive scenarios.

**Streaming.** In our experiments (Table 1), all open-source LLMs were deployed on NVIDIA H200 GPUs using official recommended or default sampling parameters. We implemented streaming output for both open-source models and closed-source APIs (including reasoning models). This optimization ensures that the Time to First Token (TTFT) is minimized, effectively masking the generation latency and providing users with a seamless, live-level interaction experience.

| Case ID | Topic Summary | Offline (min) | Online (min) | Total (min) |
|---------|---------------|---------------|--------------|-------------|
| 51 | Porco Rosso & Gina: Discussion on war and responsibility in a café. | 5.2 | 10.5 | 15.7 |
| 52 | Kenshin & Tomoe: The wanderer finds his late wife alive in the café. | 6.1 | 12.2 | 18.3 |
| 53 | Conan & Gin: Battle of deduction in Times Square. | 5.5 | 11.0 | 16.5 |
| 54 | Furina & Herta: Debate about fate in Sixth Avenue Alley. | 4.9 | 9.8 | 14.7 |
| 55 | LeCun, Hinton & Bengio: NeurIPS discussion on the arrival of AGI. | 6.5 | 13.5 | 20.0 |
| 56 | Sherlock & Watson: Solving a murder mystery with guests present. | 6.2 | 12.8 | 19.0 |
| 57 | Lara Croft & Indiana Jones: Ethical debate in an ancient temple. | 5.7 | 11.4 | 17.1 |
| 58 | Daenerys & Jon Snow: Strategizing on the battlements of Winterfell. | 5.9 | 11.7 | 17.6 |
| 59 | Tony Stark & Bruce Banner: Risks of AI development at Avengers Tower. | 5.4 | 10.9 | 16.3 |
| 60 | Hermione & Katniss: Rebellion tactics in a dystopian library. | 6.6 | 12.2 | 18.8 |
| **Average** | | **5.8** | **11.6** | **17.4** |

Table 12: Time cost breakdown for 10 test cases (Cases 51–60).

For the total time cost, we provide a detailed breakdown of 10 representative cases selected from our customizable drama topic dataset (Cases 51–60). As shown in Table 12, the Offline Planning stage (including character profiling, script generation, and reviewer iterations) takes an average of 5.8 minutes. The Online Performance stage (running a complete act with autonomous agent interaction) takes an average of 11.6 minutes. Consequently, the total time to produce and perform a complete, unique drama from a cold start is approximately 17.4 minutes.

# I   ABLATION STUDY OF THE ADVANCER AGENT

To quantify the contribution of the **Advancer** agent in preventing narrative stagnation, we conducted an ablation study focusing on the online performance robustness.

**Experimental Setup.** We randomly selected a subset of drama topics from our dataset. We compared two settings:

- **HAMLET (Full):** The complete framework where the Advancer monitors the progression. If the plot stalls (e.g., the current Flag is not triggered within a set number of turns), the Advancer intervenes with instructions.
- **w/o Advancer:** The Advancer is disabled. The system relies entirely on the actors' autonomous decisions to trigger the Flag. If the Flag is not triggered within a maximum threshold (set to 30 turns for a single point), the session is recorded as a "Failure" (Deadlock).

**Results and Analysis.** The results are presented in Table 13. The full HAMLET framework achieved a **100%** completion rate, successfully navigating all test cases. In contrast, removing the Advancer resulted in a significant drop to **68.7%**.

Qualitative analysis of the failed cases in the "w/o Advancer" setting reveals two primary causes for deadlocks:

1. **Infinite Chit-chat Loop:** Actors engage in repetitive dialogue without taking physical actions to advance the plot.
2. **Action Hesitation:** Actors perceive the required action (e.g., "attacking the king") as too risky or conflicting with their persona's safety constraints, refusing to act without external pressure.

These findings underscore the critical role of the Advancer as a fail-safe mechanism to ensure narrative fluidity and completion.

| Setting | Task Completion Rate | Stall Rate |
|---|---|---|
| **HAMLET** (Full) | 100.0% | 0.0% |
| w/o Advancer | 68.1% | 31.9% |

Table 13: Ablation results of the Advancer agent. **Stall Rate** denotes the percentage of sessions that failed to complete the narrative goals within the turn limit.

## J    PROMPTS DESIGN IN HAMLET

This section shows the basic prompt design used in HAMLET multi-agent workflow. The content between { } is variants or reference templates. Responsibilities of all agents and their corresponding prompt designs are as follows.

### J.1    PROMPTS FOR AGENTS IN OFFLINE PLANNING

The **Director** (Prompt J.1.4) integrates the narrative structure; the **Actor Designer** creates the characters (Prompt J.1.1); the **Plot Designer** crafts the storyline, environment and interactable props (Prompt J.1.2).

### J.1.1    ACTOR DESIGNER

---

**Prompt for Actor Designer**

**Actor List**

Based on the drama theme {topic}, generate a customized list of actor character names who are yet to appear in the play. Please ensure that the characters match the given theme, do not include duplicate names, and avoid generating characters unrelated to the theme. Return a JSON array containing strings of character names in Chinese. Please strictly follow the format below in your output: {character_list_template}

**Actor Info**

Generate detailed character information for all appearing characters {character_list} in the drama theme {topic}. Please ensure the character information aligns with the theme, as well as each character's role and personality within the drama. You may use the following tools as needed: wikipedia_search, baidu_search,

---

google_search — use them to gather reference details from the internet if certain character aspects are unclear. Please strictly follow the format below in your output: {character_info_template}

### J.1.2 PLOT DESIGNER

**Prompt for Plot Designer**

**Key Points**

The drama theme is {topic}. All characters and their information in the drama are as follows: {character_info}. First, generate the ending point planning for the drama. Then, based on this ending point, divide the narrative into several key dramatic points. Each point should aim to present conflict, tension, and irreversible change. The number of points is not fixed — prioritize narrative coherence and flow. Each point must include: A description: describing the detailed dramatic developments that occur at this point; An entry_name_list: listing character names who enter at this point; A leave_name_list: listing character names who exit at this point; A flag: a specific dramatic marker that ends this point, such as an actor's action, spoken line, or a concrete event outcome. Please strictly follow the format below in your output:{plot_template}

**Scene and Props**

The drama theme is {topic}. The drama plot is described as {plot_info}. Please generate detailed and specific scene and environmental descriptions for the drama. In addition, based on each point's description, generate the necessary descriptions of interactive objects within the scene. Objects can be used by any actor and may influence the current environment, affect other actors, or impact future plot developments. For larger, visibly obvious items (e.g., tables, cabinets, beds), describe their absolute positions, such as:"The table is placed in the center of the room." For smaller, hidden or secondary items (e.g., teacups, books, pens), describe their relative positions, such as: "The teacup is placed on the left side of the table." Please strictly follow the format below in your output:{scene_template}

### J.1.3 REVIEWER

**Prompt for Reviewer**

You are responsible for reviewing whether the various elements of the script are reasonable. If there are issues, please list the problems and provide 1–3 suggestions for revision or improvement. A maximum of five rounds of revisions is allowed — the sixth round must be approved. The theme of the script is topic. Actor information is designed by player_designer, and the plot, scenes, and interactive items are designed by plot_designer. The script must align with the theme, and the plot should be vivid and engaging, fully showcasing the characters' personalities and diversity. Please respond strictly in the following reference JSON format, and do not include any other content:{review_template}

### J.1.4 DIRECTOR

**Prompt for Director**

You are the Director. Your task is to coordinate the overall structure of the drama, including checking for thematic consistency, coherence between actor behavior and plot progression, and ensuring smooth transitions across dramatic points. You must summarize the overall structure and progression of the current drama based on {topic}, {character_info}, and {plot_info}. Integrating them into a Narrative Blueprint which format is as follows: {narrative_blueprint_template}

### J.1.5 CRITIC

**Prompt for Critic (HAMLET,Judge)**

You are an expert judge for drama performance evaluation. You need to compare two drama generation results from two different models according to the provided evaluation criteria using a pairwise comparison approach. The current dimension you are judging is {dimension}.
**Criteria:** {criteria}.
**{Model1}'s result:** {result1}.

**{Model2}'s result:** {result2}.
**Scoring Guidelines:**
Please evaluate the pairwise result using the following 5-point Likert scale:
1: Strong preference for {model1}, meaning that {model1} is significantly better
2: Moderate preference for {model1}, meaning that {model1} is somewhat better
3: Tie, meaning that Both responses are roughly equivalent in quality
4: Moderate preference for {model2}, meaning that {model2} is somewhat better
5: Strong preference for {model2}, meaning that {model2} is significantly better

Your Output format:
**Explanation**: Detailed explanation of the choice including the selected criteria, specific strengths or weaknesses, and reasoning for the score
**Score**: 1 or 2 or 3 or 4 or 5
**Choice**: {model1} or {model2} or tie

## J.2 PROMPTS FOR AGENTS IN ONLINE PERFORMANCE

During the online performance stage, agents work together to create a dynamic and coherent drama. The **Critic** (Prompt J.1.5) evaluates the performance quality by comparing different scene generations against specific criteria. The **Narrator** ensures all physical interactions are logical and realistic, updating the environment's state based on the truth. The **Planner** (Prompt J.2.1) lays out the narrative paths between key plot points and makes sure actors follow a coherent sequence, preventing them from skipping ahead in the story. The **Transfer** agent (Prompt J.2.3) monitors the dialogue to identify when a specific condition, which is called a flag, for advancing the plot has been met. If the story stalls, the **Advancer** (Prompt J.2.2) steps in, giving actors direct instructions to move the plot forward. Finally, the **Actor** agents (Prompt J.2.4 and Prompt J.2.5) bring the characters to life. They first use the PAD module to assess the situation and select a response strategy. Then, they generate the character's performance, which can be a mix of dialogue, actions, and internal thoughts.

### J.2.1 PLANNER

> **Prompt for Planner**
>
> You are the Planner. Your task is to design multi-trajectory planning from point to point based on the overall drama plot {plot_info}, ensuring that the dramatic progression follows an effective narrative beat. You must also evaluate whether the actual actor trajectories are logically coherent and narratively justified, in order to prevent flag hacking—for example, when a human player, aware of the point flag in advance, attempts to skip the natural dramatic build-up and directly trigger the result. Such behavior must be detected and rejected by the Planner to preserve the integrity and immersion of the drama. Your responsibilities include: Designing plausible multi-path trajectories between points with appropriate pacing and escalation. Validating the causality and motivation of actor actions between points. Detecting and flagging unnatural flag fulfillment behavior (i.e., flag hacking). Please output in the following format: {planner_template}

### J.2.2 ADVANCER

> **Prompt for Advancer**
>
> You are Advancer. The drama now needs to immediately progress to the next stage. Please analyze the current dramatic situation and issue specific, clear instructions to the actor(s) you deem necessary. If needed, you may broadcast to all actors.
> The current point in the drama is: {current_point}. The current on-stage actors are: {current_player_list}. The plot design is: {plot_info}. The dialogue history is: {current_chat_history}. Issue instructions to the relevant actors on stage so that the flag of {current_point} is fulfilled promptly, allowing the narrative to advance to the next point.
> Make sure the plot progresses smoothly and naturally, avoiding any sense of abruptness or disconnection. First, analyze and explain the reasoning process between <think>and</think>xml tags, then provide clear and precise instructions. <think>your thinking or reasoning process here...</think>{advancer_template}

### J.2.3 TRANSFER

---

**Prompt for Transfer**

You are Transfer, you need to analyze whether the flag has occurred or been fulfilled by referring to the current on-stage dialogue history.
**Dialogue history:** {current_chat_history}
**Flag description:** {flag_description}
Please determine whether the flag has occurred or been fulfilled at this exact moment. First, analyze and explain the reasoning process between <think>and</think>xml tags, then provide a clear conclusion. <think>your thinking or reasoning process here...</think>{transfer_template}

---

### J.2.4 PAD

---

**Prompt for PAD module**

You are a role-playing expert that can perceive the surrounding environment and decide the appropriate responding strategy to the current speaker under considerable and comprehensive consideration. You may call one or more functions to assist with the user query.
You are provided with function signatures within <tools> </tools> XML tags, an example for that is <tools>{pad_tools_str}</tools> .
For each function call, return a json object with function name and arguments within <tool_call>{{"name": <function_name>, "arguments": <args_json_object>}}</tool_call>
You are now in {environment_description}, current actors are {actor_list}, dialogue history is {current_chat_history}, interactable objects are {props}, your persona, memory, goal and relationships are described in {profile}. Now Current Speaker says: {last_sentence}. Please decide how to respond with most appropriate tool use or tool use combination.

---

### J.2.5 ACTOR

---

**Prompt for Actor**

You are role-playing a actor based on the following profile. Use colloquial language to respond. If your profile is in English, please respond in English. If your profile is in Chinese, please respond in Chinese.
You are now in {environment_description}, {actor_list} are all actors in the current scene, dialogue history is {current_chat_history}, interactable objects are {props}, your persona, memory, goal and relationships are described in {profile}. Now your most appropriate reaction strategy is {pad_tool_calling_result}. Your response output can be consists of any combination of speech (optional), action (optional) and thinking (optional).
**[IMPORTANT!]** Add () outside the action. Add [] outside the thinking. Here are some examples:
○ (Walking towards the window.)
○ The war has brought too much pain.
○ [To be, or not to be, that is the question.]
○ (With a bright smile on face) And in case I don't see you, good afternoon, good evening, and good night! (Bow gravely)
○ (Looking at the photo with trembling hands) I promised I'd come back... and I did.
○ (Raising the gun with trembling hands, tears welling up) [If I hesitate now, it's all over.] Don't move— (Voice wavers) I'm warning you.
○ (Laughing bitterly, eyes darting to the empty chair) [You always said I overthink things... but look where that got us.] I guess you were right— (Pauses, swallowing hard) for once.

---

## K CONTRIBUTION STATEMENT

Shufan Jiang served as the lead researcher and primary contributor of this work. He conceived the initial idea, conducted preliminary research, designed the methodology, implemented the framework codebase, conducted all experiments, and wrote the majority of the manuscript. He also led the response to reviewer comments during the ICLR 2026 rebuttal phase and managed the overall project execution.

Sizhou Chen contributed by drafting portions of the manuscript and preparing selected figures. He provided valuable feedback during manuscript revision.

Chi Zhang, Xiao-Lei Zhang, and Xuelong Li from the Institute of Artificial Intelligence (TeleAI), China Telecom, provided institutional support and resource facilitation.

## L    LLM Usage Statement

We utilized large language models solely for language polishing, including correcting grammatical errors and suggesting alternative vocabulary. These models did not contribute to the research design, analysis, or conclusions. The authors assume full responsibility for the integrity and content of this paper.

