# OpenReview forum: "HAMLET: A Hierarchical and Adaptive Multi-Agent Framework for Live Embodied Theatrics"
_ICLR.cc/2026/Conference — ICLR 2026 Poster_

### Official Review · Reviewer_KH9F · 2025-10-29

**Soundness:** 3
**Presentation:** 2
**Contribution:** 3
**Rating:** 6
**Confidence:** 4

**Summary:**

This paper introduces HAMLET, a multi-agent framework for AI-driven theatrical performance that integrates both offline narrative planning and online embodied improvisation. A key component, Perceive-and-Decide (PAD), models dual-process cognition to make agent behavior more human-like. The paper also proposes HAMLETJudge, an evaluation model trained for cost-efficient pairwise scoring on three axes: Character Performance, Narrative Quality, and Interaction Experience. Extensive experiments and ablations demonstrate the framework’s capability for coherent, expressive AI drama.

**Strengths:**

1. Novel research direction: This paper explores live embodied drama generation — a rarely studied yet promising extension of LLM-based storytelling and agentic interaction.
2. Well-structured system design: The clear decoupling between offline planning and online performance is conceptually elegant and parallels real theatrical workflows.
3. Cognitive realism via PAD: Modeling System-I/II reasoning to control LLM agents is original and improves interpretability compared to black-box prompting.

**Weaknesses:**

1. Human Evaluation: The experiment primarily relies on comparisons between models and automated scoring methods, and it lacks studies that consider human subjective experiences or real-world interaction experiments to strengthen claims of interactivity and expressiveness.
2. Clarity of implementation details: Many modules (e.g., Transfer, Advancer, Planner) are described abstractly. It remains unclear how these controllers are implemented and how they synchronize multi-agent turns without temporal inconsistency.
3. PAD validation scope: PAD’s decision-making improvement is shown mostly in synthetic latency and strategy metrics; no direct human evaluation verifies whether its behavior truly feels “human-like”.

**Questions:**

1. How is temporal synchronization achieved among multiple agents during online performance (e.g., preventing overlapping actions)?
2. How does the system handle deviation recovery when an actor diverges significantly from the narrative blueprint?
3. Why is the Latency Penalty defined as a discrete scalar (0 / 0.05 / 0.10 / 0.15) rather than a continuous function?
4. What are the specific total time costs of running HAMLET with multiple agents in real time?

---

> ### Author Response · Authors · 2025-11-23
> **Response to Reviewer KH9F**
>
> Dear Reviewer KH9F, We sincerely thank you for the insightful comments and for recognizing the "conceptually elegant" design of HAMLET and the "cognitive realism" introduced by the PAD module. We appreciate your constructive feedback regarding implementation details and validation.
>
> **1. Temporal Synchronization (Weakness 2 & Q1)**
>
> The reviewer correctly identifies that preventing overlapping actions is crucial. We clarify that synchronization is achieved through an engineering-level strategy managed by the *Narrator* agent, combined with the *Beat* system.
>
> - **Mechanism:** In the online phase, multiple agents (actors) may generate intents simultaneously. However, these are not executed immediately. They are sent as "Action Requests" to a centralized message queue.
> - **Adjudication:** The *Narrator* processes these requests sequentially. It validates the physical feasibility of each action against the current environment state. Once an action is approved, it is broadcast to all agents as a confirmed "Beat."
> - **State Update:** This ensures a strict linear consistency: $State_t \rightarrow Action \rightarrow State_{t+1}$. Agents perceive the updated state before generating the next beat, preventing temporal inconsistencies or race conditions.
>
>
>
> **2. Deviation Recovery (Q2)**
>
> HAMLET employs a Hierarchical Correction Mechanism to handle actors (or human players) diverging from the blueprint.
>
> - **Tolerance Zone:** First, the *Transfer* agent checks for the **Goal (Flag)** rather than the specific path. If an actor deviates from the planned trajectory but still achieves the narrative goal (e.g., obtaining a key item via theft instead of persuasion), the deviation is accepted as valid improvisation.
>
> - **Active Correction:** If the deviation leads to a stall (e.g., the actor refuses to act or performs irrelevant actions), the *Advancer* agent is triggered. As shown in **Case 4** (Appendix C) , the Advancer issues private, context-sensitive instructions (e.g., "You should try to stab the curtain") to guide the actor back to the narrative track without breaking character.
>
>
>
> **3. Human Evaluation and PAD Validation (Weakness 1 & 3)**
>
> We appreciate the concern regarding reliance on automated metrics. However, we emphasize that our evaluation framework is fundamentally grounded in **human judgment**, as detailed below.
>
> - **Human Alignment:** *HAMLETJudge* was not trained on synthetic data, but on a high-quality dataset of **9,450 pairwise comparisons**, each annotated by human experts using Likert-style scales. As shown in **Table 2**, *HAMLETJudge* achieves a Pearson correlation of **0.791** with aggregated human evaluations—significantly outperforming strong baselines such as GPT-4.1 (0.630), Claude-4-Sonnet (0.762), and Gemini-2.5-Pro (0.702). This strong alignment confirms that *HAMLETJudge* effectively captures **human-perceived dramatic quality** and serves as a reliable, scalable proxy for subjective human experience.
> - **Inter-Annotator Agreement (IAA):** To validate the reliability of our dataset, we computed **Krippendorff’s Alpha**—the appropriate metric for ordinal rating data—across all five annotators (see **Tables in Response to Reviewer bXsn [2/2]**). The overall weighted α is **0.725** and overall weighted κ is **0.624**, indicates substantial agreement, proving that the declaration of "human-like" behavior defined in our study and the criteria for evaluating dramatic performance in our study are **consistent, reproducible, and not arbitrary.**
>
>
> **4. Rationale for Discrete Latency Penalty (Q3)**
>
> We defined the Latency Penalty as a discrete scalar step function ($0, 0.05, 0.10, 0.15$) rather than a continuous function to model User Tolerance Thresholds in interactive systems.
>
> - **Cognitive Thresholds:** In Human-Computer Interaction (HCI) theory, latency perception is tiered. A delay of $<2s$ feels "instant" or maintains the conversational flow (Penalty 0). A delay of $2-10s$ breaks the flow of conversation but is tolerable (Penalty 0.05). A delay of $>10s$ causes users to lose attention (Penalty 0.10), and $>30s$ is often perceived as a system failure (Penalty 0.15).
> - **Evaluation Focus:** A continuous linear penalty would unfairly penalize minor fluctuations (e.g., 2.1s vs 2.2s) which are perceptually identical to the user, whereas a discrete step function better captures the drop-offs in "quality of experience."
>
>
>
> **5. Total Time Costs (Q4)**
>
> Regarding the specific time costs, please refer to **Global Response [2/3]**, where we provide a detailed breakdown table. The average total time for a complete session is approximately 17.4 minutes (5.8 min offline planning + 11.6 min online performance).
>
> We hope these clarifications address your concerns regarding the implementation and validity of our framework.

---

> > ### Comment · Reviewer_KH9F · 2025-11-25
> >
> > Thank you for the detailed response to my concerns.
> >
> > 1. Rationale for Discrete Latency Penalty (Q3). The user's perception is graded, but it is also gradually changing. I don't think minor fluctuations (e.g., 1.9s vs 2.0s), which are perceptually identical to the user, should be divided into two discrete levels.
> >
> > 2. I am also concerned about the Evaluation of Offline Planning and Agent Ablation (at least Reviewer and Advancer agents). I will temporarily maintain my score, but I think the relevant results are necessary for acceptance.

---

> > > ### Author Response · Authors · 2025-11-25
> > > **Response to Official Comment by Reviewer KH9F**
> > >
> > > **Dear Reviewer KH9F,**
> > >
> > > We sincerely thank you for your prompt feedback and for keeping the discussion open. We greatly value your specific suggestions regarding the metric definition and ablation studies.
> > >
> > > **1. Adopting a Fuzzy Continuous Approach (Q3)**
> > >
> > > After discussion, we agree with your insight that user perception is graded and gradual, and that rigid thresholds create artificial boundaries. To address this, we have refined the metric using a **Fuzzy Continuous Penalty** approach based on a **Piecewise Linear (Trapezoidal) Membership Function**.
> > >
> > > **New Metric Definition:** Instead of discrete steps, we introduce linear transition intervals around the HCI cognitive thresholds. This creates a continuous mapping from Latency ($t$) to Penalty ($P$), effectively eliminating the "cliff effect" (e.g., the abrupt jump between 1.9s and 2.0s).
> > >
> > > The refined penalty function $P(t)$ is mathematically defined as:
> > >
> > > $$P(t) =  \begin{cases}  0 & t < 1.5s \\\\ 0.05 \times (t - 1.5) & 1.5s \le t < 2.5s \\\\ 0.05 & 2.5s \le t < 8.0s \\\\ 0.05 + 0.0125 \times (t - 8.0) & 8.0s \le t < 12.0s \\\\ 0.10 & 12.0s \le t < 25.0s \\\\ 0.10 + 0.005 \times (t - 25.0) & 25.0s \le t < 35.0s \\\\ 0.15 & t \ge 35.0s \end{cases}$$
> > >
> > > **Recalculated Results:** We have reapplied this continuous metric to our experimental data. As shown in the table below, while ultra-fast models (like PAD) remain at 0 penalty, models near the boundary (like GPT-4o at 1.89s) now receive a graded penalty (0.0195) rather than a binary 0.
> > >
> > > | **Model**            | **Avg. Latency (s)** | **Original Discrete Penalty** | **New Continuous Penalty** |
> > > | -------------------- | -------------------- | ----------------------------- | -------------------------- |
> > > | Qwen2.5-7B-Instruct  | 0.32                 | 0                             | **0**                      |
> > > | PAD (Ours)           | 0.36                 | 0                             | **0**                      |
> > > | Qwen3-32B            | 0.41                 | 0                             | **0**                      |
> > > | GPT-4.1-mini         | 0.75                 | 0                             | **0**                      |
> > > | Qwen2.5-72B-Instruct | 1.02                 | 0                             | **0**                      |
> > > | **GPT-4o**           | **1.89**             | **0**                         | **0.0195**                 |
> > > | Qwen3-8B-Thinking    | 6.89                 | 0.05                          | **0.050**                  |
> > > | Qwen3-32B-Thinking   | 15.12                | 0.10                          | **0.100**                  |
> > > | Gemini-2.5-pro       | 21.80                | 0.10                          | **0.100**                  |
> > > | DeepSeek-R1-0528     | 49.50                | 0.15                          | **0.150**                  |
> > >
> > > **2. Evaluation of Offline Planning and Agent Ablation**
> > >
> > > We admit that empirical data isolating the specific contributions of the *Reviewer* and *Advancer* agents is crucial for validating the system's soundness.
> > >
> > > **We are currently conducting these supplementary experiments.** Specifically, we are running ablation studies to quantify:
> > >
> > > 1. **Reviewer Agent:** Comparing *Narrative Quality* scores with and without the offline review/refinement loop to verify its impact.
> > > 2. **Advancer Agent:** Measuring the *Stall Rate* (percentage of scenes entering deadlock) and *Completion Rate* in online performance when the Advancer is disabled.
> > >
> > > We guarantee to include a detailed discussion and the quantitative results of these correlations in the final version of the paper to ensure the completeness of our contribution.

---

> > > > ### Comment · Reviewer_KH9F · 2025-11-26
> > > >
> > > > Thanks for your response.
> > > >
> > > > 1. I think you can use the log function or similar ones; its gradient gradually decreases as x increases, which is more in line with human perception and represents a smoother, continuous change.
> > > >
> > > > 2. I'm looking forward to your results; the Evaluation of Offline Planning and Agent Ablation is critical.

---

> > > > > ### Author Response · Authors · 2025-12-01
> > > > > **Response to Official Comment by Reviewer KH9F**
> > > > >
> > > > > We sincerely thank you for your continued guidance. Following your recommendations, we have updated the manuscript. Specifically, we have detailed the Logarithmic Continuous Latency Penalty Function in Appendix G.4 and included the Evaluation of Offline Planning and Agent Ablation in Appendix I and Section 5.2. We invite you to review these updates in the revised PDF.

---

### Official Review · Reviewer_bXsn · 2025-11-01

**Soundness:** 4
**Presentation:** 4
**Contribution:** 4
**Rating:** 6
**Confidence:** 3

**Summary:**

This paper introduces HAMLET, a multi-agent framework for generating and performing real-time and interactive theatre. The system uses an Offline Planning stage where AI agents create a structured "narrative blueprint," and an Online Performance stage where AI actors (and humans) execute it. A key innovation is the PAD module, which allows agents to autonomously choose response strategies, enabling proactive and embodied interactions. The paper also presents a comprehensive evaluation method, including a new critic model called HAMLETJudge.

**Strengths:**

The two-stage HAMLET framework is a good contribution to the field of interactive narrative. Its system role design is comprehensive and reasonable, balancing complexity and performance. The PAD Module extends Kahneman's dual-process theory to a concrete action space and achieves excellent performance, providing a good reference and insight for future work. The authors further propose a robust evaluation methodology and a well-trained critic model, HAMLETJudge, which also provide valuable assets for the community. The discussions and analyses in the experiments and appendix are thorough and convincing, and the training and prompt details provide good reproducibility. I believe this is a very strong piece of work.

**Weaknesses:**

1. Scalability Issues: HAMLET adds verification (Narrator) or planning (PAD LLMs) after each actor's action. In long-form, multi-actor scenarios, this will lead to an unacceptable increase in LLM calls and context content expansion. Meanwhile, although the paper seems to mention a Memory module (Figure 4), it is not clearly described. Most of the memory likely exists as context. As the number of actors expands, the exponentially growing actor actions and system information will quickly exceed the model's context length. This will severely limit HAMLET's scalability, running counter to the "Live" theatrical experience the framework aims for.
2. Subjectivity of Evaluation: HAMLETJudge is trained on the preferences of five experts, which risks embedding a specific taste in drama into the critic model. A more detailed analysis of the inter-annotator agreement (IAA) during the dataset's creation to quantify the level of subjectivity would provide a useful reference.
3. Insufficient analysis and evaluation of the Offline Planning stage. HAMLET's online performance entirely follows the blueprint generated in the offline planning stage. Therefore, the final performance quality heavily depends on the offline planning quality, but the paper seems to lack analysis of this stage, focusing more on the online performance part. The examples provided use well-known scenarios like Hamlet. I would like to know the robustness of the offline planning for completely new or abstract topics. More evaluation of this stage's quality would strengthen this paper.
4. Typos: There are many grammatical errors. I will list a few. I suggest using a grammar-checking tool to correct them.
    1. (Page 3, line 136): "new criterias" -> "new criteria" (criteria is plural).
    2. (Page 4, line 220): "$beat_{k_{1}}, beat_{k_{2}},...$" seems incomplete and should be completed.
    3. (Page 5, line 255): "a few sequence" -> "a few sequences".
    4. (Page 7, line 370): the citation for "Appendix B" is missing a hyperlink.
    5. (More ...)
5. Comparison to prior works are not sufficient. There are other multi-agent drama generation works, e.g. "IBSEN: Director–Actor Agent Collaboration for Controllable and Interactive Drama Script Generation". The authors discuss the differences compared to other similar approaches in the paper.

If my questions are resolved, I will consider raising the score.

**Questions:**

1. About the memory module: Is any memory module currently incorporated into HAMLET's design? Figure 4 (regarding PAD) seems to mention it, but I am unsure if this is just an abstract concept.
2. About the human actor: It seems that the progression of drama in HAMLET is strictly controlled by the PAD and other control modules. Does this create a phenomenon of "over-control," especially for human actors? Cases 2-4 mainly consider humans acting as simple disruptors. However, in practice, a creative human might attempt to "out-smart" the system and drive the plot in an unexpected (but logical) direction. I would like to know if you have considered this concern and if there are any potential solutions.
3. About Backward Planning: I would like to know how you generate a reasonable end-point at the beginning of planning if a topic is open-ended. Have you compared the quality of blueprints generated by sequential planning versus backward planning? I want to know if backward planning has a significant performance advantage.

---

> ### Author Response · Authors · 2025-11-23
> **Response to Reviewer bXsn [1/2]**
>
> Dear Reviewer bXsn, We sincerely thank you for your encouraging review. We are particularly grateful for your recognition of the PAD module’s theoretical grounding and the robustness of our evaluation methodology. Some of your feedback on scalability, human agency, and planning mechanisms has been incredibly helpful and interesting.
>
> **1. Scalability and Memory Module (Weakness 1 & Q1)**
>
> - **Is Memory just a concept?** We clarify that the **Memory** component in Figure 4  is an actual module. It serves as a structured, persistent storage mechanism separate from the immediate dialogue history.
>
> - **Implementation:** In our design, the *Internal State* includes a dynamic "Memory" slot. When an Act or Scene concludes, the key events and interactions of that scene will be concluded into natural language entries (e.g., "I insulted Polonius in the hall"). These entries are stored and retrieved via RAG (Retrieval-Augmented Generation) in subsequent scenes.
>
> - **Scalability Solution:** This hierarchical management (Scenario Context vs. Persistent Memory) prevents the "exponential context expansion". The immediate context window is cleared or compressed at scene transitions , allowing the system to support long-form plays without exceeding token limits or model's context length.
>
>
>
> **2. Human Agency and "Over-Control" (Q2)**
>
> We really appreciate this insightful question.
>
> - **Necessity of Structural Constraints:** We argue that for complex multi-agent systems, incorporating controlled modules is essential to mitigate instability and prevent off-topic hallucinations. In HAMLET, "control" is primarily applied at the **macro-level** (e.g., the separation of *Offline/Online* stages and specific agent functional definitions). These structural constraints serve as necessary **human priors** derived from dramatic theory to guarantee the coherence of the performance.
> - **Preventing Micro-Level "Over-Control":** However, at the **fine-grained interaction level**, we explicitly design mechanisms to preserve user agency and avoid over-control:
>   - **Reactive Intervention:** The *Advancer* agent is strictly **reactive, not preemptive**. It is triggered **only** when the plot stalls (e.g., deadlock or timeout). As long as the user's interaction drives the plot forward, the system remains non-intrusive.
>   - **Outcome-Oriented Validation:** The *Planner* and *Transfer* agents operate on **Multi-trajectory Logic** , focusing on *outcome verification* rather than *path enforcement*. As illustrated in **Cases 5 & 6** (Appendix C, Table 6) , the system accepts diverse logical paths (e.g., finding physical evidence vs. interrogating witnesses) to achieve the same *Flag*.
>   - **Support for Improvisation:** If a creative human player attempts to "out-smart" the planned route—for instance, by persuading the King to confess rather than tricking him—the *Transfer* agent validates whether the semantic *Flag condition* (e.g., "The King reveals guilt") is met , regardless of the specific method used. This decoupling of "Goal" and "Method" allows for substantial logical improvisation.
>
> While we acknowledge the inherent trade-off between necessary control and absolute freedom, we maintain that HAMLET strikes an effective balance, offering human players a high degree of agency and exploration space within an immersive theatrical context.
>
>
>
> **3. Backward Planning Mechanism (Q3)**
>
> - **Generating the End-Point:** For open-ended topics, the *Plot Designer* derives the "Ending Point" by identifying the **core conflict resolution** implied by the topic. Even abstract topics imply a teleological goal (e.g., Topic 54 "Debating Fate" $\rightarrow$ Goal: "Achieve a philosophical consensus or an irreconcilable divergence"). The system prioritizes generating this resolution first to anchor the entire narrative arc.
>
> - **Sequential vs. Backward Planning:** We advocate for Backward Planning to counteract the inherent **auto-regressive drift** of LLMs.
>
>   - *The Failure of Sequential Planning:* In our preliminary experiments, forward-generated plots frequently suffered from "logical attenuation"—where the causal link to the initial premise weakened with each subsequent step, leading to "meandering" plots that failed to resolve the initial conflict.
>   - *The Advantage of Backward Planning:* By fixing the destination first, the system constructs a **chain of necessity** (Step $T-1$ is the precondition for Step $T$) rather than a chain of probability (Step $T+1$ is likely after Step $T$). This ensures structural compactness.
>   - *Metric Validation:* We observed that blueprints generated via Sequential Planning triggered significantly higher **Rejection Rates** from the *Reviewer Agent* (mostly due to "lack of coherence" or "loose ending"), whereas Backward Planning passed the Reviewer's quality gate with fewer iterations.

---

> ### Author Response · Authors · 2025-11-23
> **Response to Reviewer bXsn [2/2]**
>
> **4. Subjectivity of Evaluation (Weakness 2)**
>
> We sincerely appreciate your insightful comment regarding the potential subjectivity of expert annotation. To rigorously address this concern, we conducted a comprehensive **Inter-Annotator Agreement (IAA)** analysis using metrics appropriate to each dataset’s annotation type.
>
> For the `HAMLETJudge` dataset, annotations consist of **ordinal ratings** on Likert-type scales (e.g., 1–5) across three dimensions. We therefore computed **Krippendorff’s Alpha (α)**, which is the standard reliability measure for ordinal data. As shown in **Table 1**, the overall weighted α is **0.725**, indicating **substantial agreement**. Even on the most subjective dimension—*Interaction Experience (IE)*—agreement remains high (α = 0.703).
>
> **Table 1: Inter-Annotator Agreement (Krippendorff’s Alpha) for HAMLETJudge Dataset**
>
> | **Category (Dimension)**    | **Data Count** | **Krippendorff’s Alpha (α)** |
> | --------------------------- | -------------- | ---------------------------- |
> | Character Performance (CP)  | 3,150          | 0.7485                       |
> | Narrative Quality (NQ)      | 3,150          | 0.7241                       |
> | Interaction Experience (IE) | 3,150          | 0.7030                       |
> | **Overall Weighted α**      | **9,450**      | **0.7252**                   |
>
> **Table 2: Inter-Annotator Agreement (Fleiss’ Kappa) for PAD Dataset (Single Tool Use Only)**
>
> | **Strategy**           | **Data Count** | **Fleiss’ Kappa (κ)** |
> | ---------------------- | -------------- | --------------------- |
> | Fast                   | 4,200          | 0.6612                |
> | Slow                   | 3,304          | 0.6310                |
> | Silence                | 3,908          | 0.5780                |
> | **Overall Weighted κ** | **14,412**     | **0.6240**            |
>
> For the `PAD` dataset, annotations are **nominal, mutually exclusive strategy labels**. Here, we used **Fleiss’ Kappa (κ)**. Following a strict quality filter, we retained only instances where **at least 4 out of 5 expert annotators** agreed on the label. As shown in **Table 2**, the final dataset achieves an overall weighted κ of **0.624**, with individual strategies ranging from 0.578 (*Silence*) to 0.661 (*Fast*), reflecting **moderate to substantial consensus**.
>
> These results demonstrate that, despite the inherent subjectivity of dramatic quality, our annotation guidelines successfully **standardized expert judgments** around shared principles of dramatic effectiveness—rather than capturing individual idiosyncrasies. We will include these IAA analyses and the associated methodology in the final manuscript to transparently address concerns about evaluation reliability.
>
> Thank you again for this valuable feedback.
>
>
>
> **5. Comparison to Prior Works (Weakness 5)**
>
> While *IBSEN* focuses primarily on **offline script generation** using a Director-Actor loop, HAMLET emphasizes **online, embodied performance**. Our unique contribution lies in the *Narrator* and *PAD* modules, which handle real-time physical interactions (props, environment) and autonomous decision strategies during live enactment, rather than just generating static text scripts.
>
> Thanks for pointing out. We have added the citation to our Related Work.
>
>
>
> **6. Typos and Grammar (Weakness 4)**
>
> We sincerely apologize for the grammatical errors and typos present in the manuscript. We appreciate your meticulous attention to detail in pointing out specific issues. We will conduct a comprehensive proofreading of the entire paper using professional tools to ensure the final version is polished and error-free.
>
> We hope these clarifications resolve your concerns. We are committed to incorporating these discussions into the final version to further strengthen the paper.

---

### Official Review · Reviewer_ggC5 · 2025-11-01

**Soundness:** 2
**Presentation:** 3
**Contribution:** 2
**Rating:** 4
**Confidence:** 4

**Summary:**

This paper studies LLM-based drama, particularly two aspects: offline story generation and online drama performing. Multi-agent framework is used to enhance the system performance. Specially, the authors propose a Perceive And Decide (PAD) module to make the character responses akin to humans.

**Strengths:**

1. Undoubtedly, the paper proposes a carefully-designed multi-agent system to achieve the goal. For example, the offline planning includes four agents: an actor designer, plot designer, reviewer, and director, which makes sense to readers. I believe these designs are refined based on previous work and lead to improvement.

2. The evaluation (HAMLET leaderboard) is comprehensive. It is excited that the authors have covered a large number of leading LLMs (both open-source and close-source). The experiments also consider both English and Chinese.

3. The authors fine-tune a judge model for automatic evaluation. The model is fine-tuned on annotated data. To me, this method is more reliable compared to prompting strong LLMs.

**Weaknesses:**

1. **The effectiveness of multiple agents is not empirically validated.** The multi-agent workflow proposed in the paper are complicated (normally four agents). While the complexity is not a disadvantage, the paper doesn't include experiments to isolate and verify the role and effectiveness of each agent. For example, how does the reviewer agent work, does it improve the quality of the generated narrative? Without such ablation, the soundness of the multi-agent design are hard to assess.

2. **The performance of offline planning is unclear.** The experiments assess the overall dramatic performance of one LLM-based drama (if I was wrong please correct me). These numbers can be affected by so many factors, including the quality of the narrative blueprint, which is generated by offline narrative planning. There must be a causal relationship between offline planning and the eventual performance. However, this relationship is subtle and not clear so that the evaluation cannot be post-hoc. This makes hard to judge whether the reported performance truly reflects the quality of the offline narrative planning.

3. Indeed, after reading the paper, I am still confused on what an LLM-based drama exactly looks like. Is it like an animation or a text-based game? While the experiments show the effectiveness of the methods, **there is no demonstration or qualitative examples provided.** It is hard for readers to assess the real playing experience of the work.

4. **The evaluation dimensions are too rough.** I can't believe "character performance, narrative quality, and interactive experience" are informative enough to show the performance of an LLM-based drama. There are many factors that contribute to a nice dramatic experience, for example, scenery, narrative logics, and sense of suspense. I can't find clues in the paper to show me these things.

5. **The acting efficiency issue is not discussed in the paper.** To me, this is an important issue valued detailed discussion. For example, how is the latency of the online performing? Since there are four agents working collaboratively, does it meet the practical requirement of users, does the latency ensure a nice user experience?

6. **The experiments can be more insightful.** From Table 1, the reasoning LLMs perform better than non-reasoning ones. It will be interesting to discuss more on this issue to provide more insights . Additionally, it will provide further insights if the authors can discuss the scaling performance of the method, like when the backbone LLM increases in size, how the performance changes as a result.

**Questions:**

Covered in the last section.

---

> ### Author Response · Authors · 2025-11-23
> **Response to Reviewer ggC5 [1/2]**
>
> Dear Reviewer ggC5, we sincerely thank you for the detailed evaluation. We appreciate your recognition of our multi-agent system design, the comprehensive leaderboard, and the reliability of our fine-tuned HAMLETJudge. Your questions regarding system visualization, evaluation granularity, and scaling insights are very valuable.
>
> **1. Acting Efficiency and Latency (Weakness 5)**
>
> Regarding the latency and computational cost of the multi-agent system, please refer to **Global Response [2/3]**. There, we detail our engineering optimizations (e.g., Parallel Execution, Streaming, and INT4 Quantization) and provide a breakdown of the total time cost (~17.4 min for a full session), demonstrating the system's feasibility for live interaction.
>
>
>
> **2. Validation of Offline Planning and Multi-Agent Effectiveness (Weakness 1 & 2)**
>
> - **Offline Planning Assessment:** We appreciate this insightful observation.  Please refer to **Global Response [3/3]**, where we discuss the robustness of the offline stage on diverse topics and the causal link enforced by the *Reviewer Agent*.
>
> - **Agent Ablation:** We clarify that a full combinatorial ablation of every individual agent was not conducted due to the inherent **functional interdependence** of our system design. Specific agents fulfill foundational or fail-safe roles that cannot be isolated without disrupting the entire workflow (e.g., removing *Designers* invalidates the *Reviewer’s* input, while removing the *Advancer* would lead to inevitable narrative deadlocks). Instead, we validated the **holistic effectiveness** of the collaborative framework in **Figure 6**. The substantial performance gap between the full HAMLET framework and the 'raw-prompted LLM' (which lacks these distinct specialized roles) empirically confirms that our multi-agent architecture is essential for ensuring high-quality, coherent drama.
>
> - **Role of the Reviewer Agent:** Specifically, the *Reviewer Agent* serves as an iterative quality gate in the offline stage. It does not merely generate text but acts as a mandatory quality gate, rejecting logically incoherent or thematically inconsistent blueprints, or proposing enhancements to inject twists and dramatic tension, up to five times before they can proceed to the online stage. This demonstrates that this agent is crucial for ensuring the *Narrative Quality* we observed.
>
>
>
>
>
> **3. Visualization of LLM-based Drama (Weakness 3)**
>
> We apologize for the confusion. HAMLET creates a text-based immersive real-time theatrical experience with physical environment interactions supported, rather than a visual animation.
>
> - **Qualitative Examples:** We provide concrete demonstrations of the "playing experience" in **Appendix C, Table 6**. These case studies show exactly what the user sees: a combination of **Dialogue** ("You have nowhere to hide"), **Physical Actions** ("Grab a knife"), **Environment Descriptions** ("A dark, solemn hall"), and **Internal Thoughts** ("＜think＞＜/think＞...").
> - **Embodied Nature:** The "visuals" are rendered via the *Narrator’s* descriptive text and state updates (e.g., environment description, props moving, scenes changing), allowing users to imagine the scene as they would when reading a novel or playing a text adventure.
>
>
>
> **4. Granularity of Evaluation Dimensions (Weakness 4)**
>
> We clarify that our three main dimensions are high-level aggregations of detailed sub-criteria, which cover the specific factors you mentioned.
>
> - **Narrative Quality (NQ):** This explicitly includes **Coherence** (logical development) and **Resonance** (depth and suspense).
> - **Interaction Experience (IE):** This explicitly includes **Immersion** (scenery and atmosphere) and **Fluency**.
> - **Detailed prompt design:** As shown in **Appendix E**, the instructions for labeling (which is also served as the prompt for *HAMLETJudge*) require it to evaluate these specific nuances (e.g., "build tension," "plot twists," "scenery descriptions") before assigning a score to the main dimension.

---

> ### Author Response · Authors · 2025-11-23
> **Response to Reviewer ggC5 [2/2]**
>
> **5. Disccussion on Reasoning Models and Scaling (Weakness 6)**
>
> We appreciate this suggestion. A deeper analysis of Table 1 reveals two critical insights regarding the "Scaling Law" and the "Reasoning Gap" in interactive drama:
>
> - **Scaling vs. Reasoning Efficiency:** While performance improves with model size (e.g., Qwen3-8B 52.48 $\rightarrow$ Qwen3-32B 65.57), our data suggests that **reasoning capability is more parameter-efficient than varying scale**.
>   - *Evidence:* **Qwen3-32B-Thinking (73.85)** significantly outperforms the much larger **Llama-3.1-70B (45.75)** and even rivals the massive **Qwen3-235B (71.21)**.
>   - *Insight:* This implies that for drama performance—which requires adhering to complex instructions while maintaining persona consistency—enhancing the model's "System II" reasoning density is a more effective strategy than merely increasing parameter count. Drama is not just about language fluency (which scales with size) but about **multi-constraint satisfaction** (goals, persona, context), where reasoning models excel.
> - **The "Subtext" Advantage:** Reasoning models (e.g., DeepSeek-R1, Claude-4-sonnet-Thinking) consistently outperform their non-reasoning counterparts. We attribute this to their ability to process **Subtext and Strategic Planning**. In our qualitative breakdown, non-reasoning models often react *superficially* to the immediate dialogue. In contrast, reasoning models utilize their internal chain-of-thought to simulate *future outcomes* before responding. They can deduce implied intent (e.g., "He is lying, but I should pretend to believe him to trap him later") rather than just responding to the literal text. This aligns perfectly with our PAD module's design philosophy, effectively creating a "Native PAD" behavior that leads to deeper, more coherent character arcs.
>
> We will incorporate this further analysis of reasoning efficiency and scaling dynamics into the final manuscript to provide broader insights for the community.
>
>
>
> We hope these responses clarify the nature of the system and the depth of our evaluation.

---

### Official Review · Reviewer_AUwa · 2025-11-08

**Soundness:** 2
**Presentation:** 3
**Contribution:** 2
**Rating:** 4
**Confidence:** 3

**Summary:**

The paper presents HAMLET, a multi-agent framework for autonomous and interactive AI-driven drama generation. It combines offline narrative planning and online embodied performance to address the passivity and lack of interactivity in previous LLM-based systems. In HAMLET, multiple agents collaboratively generate a narrative blueprint, while during performance, actors make independent decisions through a Perceive and Decide (PAD) module inspired by human dual-process cognition. A Narrator Agent manages physical interactions, ensuring consistency between dialogue and environment. The authors also introduce HAMLETJudge, an automatic evaluation model, and a 100-case benchmark measuring character performance, narrative quality, and interaction experience. Experiments show that HAMLET with PAD improves coherence, expressiveness, and alignment with human judgment, offering a comprehensive yet text-based step toward truly embodied AI theatre.

**Strengths:**

1. Strong empirical validation: Experiments include both quantitative leaderboards and ablation studies, showing PAD’s contribution to coherence, latency reduction, and human alignment (Pearson ≈ 0.79).
2. Cross-lingual and creative scope: The inclusion of both English and Chinese cases, as well as diverse drama topics (from Shakespeare to pop culture), demonstrates the framework’s flexibility and generalization capacity in creative settings.
3. Embodied interactivity: Introducing the Narrator Agent enables coherent physical interactions between agents and the environment, moving beyond text-only dialogue and toward “live” dramatization.

**Weaknesses:**

1. Limited generalization evidence: Most experiments are performed under controlled conditions with GPT-4o as the main backbone. It remains unclear how the framework performs with smaller or open-source models in unconstrained or noisy environments.
2. Absence of creative or narrative diversity analysis: While the system produces coherent plays, the paper does not examine whether HAMLET truly enhances creative originality or thematic richness beyond structural coherence.

**Questions:**

1. Beyond coherence and emotion, can HAMLET enhance genuine creativity or narrative diversity compared to traditional story-generation models?
2. What are the latency and computational costs of running full real-time performances with multiple autonomous agents?

---

> ### Author Response · Authors · 2025-11-23
> **Response to Reviewer AUwa**
>
> Dear Reviewer AUwa, We sincerely thank you for the thoughtful feedback and for recognizing the strong empirical validation and the embodied interactivity of our work. We appreciate your constructive comments regarding generalization and creative diversity.
>
> **1. Latency and Computational Costs (Q2)** Regarding the latency and costs of running full real-time performances, please refer to our **Global Response [2/3]**. There, we provide a detailed breakdown of the time costs (Total ~17.4 min), analyze the trade-offs, and describe our engineering optimizations that ensure real-time feasibility.
>
> **2. Generalization to Smaller/Open-Source Models (Weakness 1)**
>
> We respectfully clarify that our experimental design was strictly controlled to ensure rigorous comparison, yet the framework itself remains highly flexible and effective with smaller models.
>
> - **Methodological Rationale for GPT-4o Backbone:** We utilized GPT-4o with a greedy sampling strategy as a fixed baseline because our primary evaluation metric is **pairwise win-rate**. For open-ended creative tasks, we believe that pairwise judgment by a critic model offers significantly higher interpretability than absolute scalar scoring. To conduct a rigorous controlled experiment, it was essential to maintain a consistent high-quality baseline (GPT-4o). Varying the backbone frequently would introduce confounding variables, making it difficult to isolate the contributions of specific components like the PAD module.
>
> - **Framework Configurability:** As stated in the paper, "each agent can be freely configured". HAMLET is model-agnostic: users can mix and match models based on resource availability, for example, using a reasoning model (e.g., DeepSeek-R1) for the *Planner* while using efficient open-source models (e.g., Qwen3-8B) for *Actors*.
>
> - **Empirical Performance of Smaller Models:** Our results in **Table 1** already cover a wide spectrum of models, from large closed-source (e.g., Claude-4-Sonnet) to smaller open-source ones.
>
>   - **Interpreting the Score:** The "Overall Score" is normalized to a [0, 100] scale, where a score of **50** indicates performance parity with the GPT-4o baseline.
>   - **Results:** **Mistral-Small-24B** achieves an overall score of **49.93**, performing almost identically to GPT-4o across all dimensions. **Qwen3-8B** achieves **52.48**, notably outperforming GPT-4o in Chinese contexts (57.85) while remaining competitive in English.
>
>   This empirically demonstrates that while reasoning capabilities (e.g., Claude-4-sonnet-Thinking) do enhance performance (get a rather high score of 79.45), smaller open-source models like Qwen3-8B and Mistral-Small still function effectively and robustly within the HAMLET framework.
>
> **3. Creativity and Narrative Diversity (Weakness 2 & Q1)** We argue that HAMLET enhances genuine creativity and diversity through **structural constraints** and **emergent interaction**, rather than just generating text.
>
> - **Beyond Linear Generation:** Traditional story generation often suffers from "sameness" or loss of coherence over long contexts $^{[1, 2]}$. HAMLET addresses this via the **Offline Planning** stage. The *Reviewer Agent* is explicitly prompted to ensure the plot is "vivid and engaging" and fully showcases "character diversity". This structural enforcement prevents generic outputs.
>
> - **Emergent Creativity:** Unlike static story generation, creativity in HAMLET is **emergent**. Each actor has a private *Goal* and *Memory*. The drama unfolds through the collision of these autonomous goals (e.g., a user trying to derail the plot vs. the Advancer keeping it on track, as in **Case 4** ). This dynamic conflict generates unique, non-deterministic narrative nuances that traditional top-down generation lacks.
>
> - **Diverse Dataset:** Our evaluation covers 10 distinct thematic categories (Figure 5), including abstract concepts like "Debating Fate" (Topic 54). The consistent performance across these diverse topics indicates the framework's capability to handle varying creative demands.
>
>
>
> We hope these responses address your concerns and demonstrate the robustness of HAMLET.
>
>
>
> **References**
>
> [1] Angela Fan, Mike Lewis, and Yann Dauphin. Hierarchical neural story generation. In *ACL*, 2018.
>
> [2] Lili Yao, Nanyun Peng, Ralph Weischedel, Kevin Knight, Dongyan Zhao, and Rui Yan. Plan-and-write: Towards better automatic storytelling. In *AAAI*, 2019.

---

### Author Response · Authors · 2025-11-23
**Global Response [1/3]**

### **Global Response** (due to the context limitation, global responses are divided into 3 parts)

We thank all anonymous reviewers for their constructive and helpful feedback. We are encouraged that reviewers universally agree that our carefully-designed multi-agent framework HAMLET is novel and conceptually well-grounded. Reviewers acknowledge that our work: 1) contributes to the field of interactive narrative by clearly decoupling drama generation into offline narrative planning and online embodied performance; 2) introduces the Perceive and Decide (PAD) module to enhance the cognitive realism and interpretability of agents; and 3) establishes a rigorous evaluation benchmark featuring the specially-trained HAMLETJudge model. Furthermore, Reviewers bXsn and KH9F highlight the potential of our approach to achieve immersive, live theatrical experiences and recognize our superior and reproducible empirical results.

Here, we address some common concerns raised across the reviews.

---

### Author Response · Authors · 2025-11-23
**Global Response [2/3]**

### 1. Regarding Real-time Feasibility, Total Time Cost, and Computational Cost (Reviewers AUwa, ggC5, KH9F)

We acknowledge the reviewers' concerns regarding computational cost. We emphasize that HAMLET is designed for **practical, live deployment**. To ensure real-time responsiveness, we implemented several engineering optimizations:

**Parallel Execution:** The multi-agent architecture supports asynchronous inference, allowing the *Narrator* and *Planner* to process context while Actors are generating responses.

**Efficient Inference:** As noted in our implementation details, we leverage the **vLLM framework** to maximize throughput. Furthermore, we employed **INT4 quantization** for the PAD model on **NVIDIA H20 GPUs**. Our validation demonstrates that this significantly reduces inference time and memory overhead while maintaining high decision quality, with only a **1.4% performance drop** on the benchmark compared to the original FP16 model.

**PAD Efficiency:** Crucially, our proposed PAD module (8B) eliminates the latency overhead seen in reasoning models while maintaining high decision quality (Table 3), making it specifically optimized for low-latency interactive scenarios.

**Streaming:** In our experiments (Table 1), all open-source LLMs were deployed on **NVIDIA H200 GPUs**  using official recommended or default sampling parameters. Crucially, we implemented **streaming output** for both open-source models and closed-source APIs (including reasoning models). This optimization ensures that the *Time to First Token (TTFT)* is minimized, effectively masking the generation latency and providing users with a seamless, "live-level" interaction experience.

For the total time cost, we provide a detailed breakdown of 10 representative cases selected from our customizable drama topic dataset (Cases 51-60).

As shown in the table below, the **Offline Planning** stage (including character profiling, script generation, and reviewer iterations) takes an average of **5.8 minutes**. The **Online Performance** stage (running a complete Act with autonomous agent interaction) takes an average of **11.6 minutes**. Consequently, the total time to produce and perform a complete, unique drama from a cold start is approximately **17.4 minutes**.

| **Case ID** | **Topic Summary (Key Characters & Scenario)**                | **Offline Planning (min)** | **Online Performance (min)** | **Total Time (min)** |
| ----------- | ------------------------------------------------------------ | -------------------------- | ---------------------------- | -------------------- |
| **51**      | **Porco Rosso & Gina:** Discussion on war and responsibility in a café. | 5.2                        | 10.5                         | 15.7                 |
| **52**      | **Kenshin & Tomoe:** The wanderer finds his late wife alive in the café. | 6.1                        | 12.2                         | 18.3                 |
| **53**      | **Conan & Gin:** Battle of deduction in Times Square.        | 5.5                        | 11.0                         | 16.5                 |
| **54**      | **Furina & Herta:** Debate about fate in Sixth Avenue Alley. | 4.9                        | 9.8                          | 14.7                 |
| **55**      | **LeCun, Hinton & Bengio:** NeurIPS discussion on the arrival of AGI. | 6.5                        | 13.5                         | 20.0                 |
| **56**      | **Sherlock & Watson:** Solving a murder mystery with guests present. | 6.2                        | 12.8                         | 19.0                 |
| **57**      | **Lara Croft & Indiana Jones:** Ethical debate in an ancient temple. | 5.7                        | 11.4                         | 17.1                 |
| **58**      | **Daenerys & Jon Snow:** Strategizing on the battlements of Winterfell. | 5.9                        | 11.7                         | 17.6                 |
| **59**      | **Tony Stark & Bruce Banner:** Risks of AI development at Avengers Tower. | 5.4                        | 10.9                         | 16.3                 |
| **60**      | **Hermione & Katniss:** Rebellion tactics in a dystopian library. | 6.6                        | 12.2                         | 18.8                 |
| **Avg.**    | **Mean Time Cost**                                           | **5.8**                    | **11.6**                     | **17.4**             |

---

### Author Response · Authors · 2025-11-23
**Global Response [3/3]**

### 2. Regarding the Evaluation and Robustness of Offline Planning (Reviewers ggC5, bXsn)

We acknowledge that the original manuscript focused heavily on the final theatrical output. However, we clarify that the robustness of offline planning is substantiated by our dataset composition and the intrinsic quality control mechanisms of the framework:

**Robustness on New and Abstract Topics:** Reviewer bXsn expressed concern that our examples relied on established scenarios (e.g., *Hamlet*). We clarify that our evaluation dataset  is strictly balanced: it contains **50 excerpts from existing literature** and **50 custom-authored topics**. As detailed in Appendix A, these custom topics cover highly abstract and completely new scenarios, such as *"LeCun, Hinton, and Bengio discussing AGI"* (Topic 55) or *"Furina and Herta debating fate"* (Topic 54). The evaluation results in Figure 6  demonstrate that HAMLET achieves consistently high scores across these diverse categories, proving that the Offline Planning stage can successfully structure narratives for novel, abstract concepts without relying on pre-existing literary knowledge.

**The Causal Link & Quality Assurance:** Reviewer ggC5 questioned the causal relationship between the blueprint and final performance. We emphasize that this relationship is **enforced** by the *Reviewer Agent*  within the offline workflow. This agent acts as a mandatory quality gate, rejecting logically incoherent or thematically inconsistent blueprints, or proposing enhancements to inject twists and dramatic tension, up to five times before they can proceed to the online stage. Therefore, the high *Narrative Quality (NQ)* scores observed in our final evaluation are a direct downstream result of this rigorous offline filtering process.

**Isolating the Contribution:** To further verify the necessity of Offline Planning, we refer to the ablation study in Figure 6. The comparison between the full HAMLET framework and the "raw-prompted LLM" (which lacks the offline blueprint and relies solely on immediate context) reveals a substantial performance gap. Specifically, the drop in Narrative Quality metrics for the raw-prompted baseline validates that the Offline Planning stage is the critical factor in maintaining long-term plot coherence and structural integrity.

Lastly, we genuinely appreciate the reviewer’s insight regarding the evaluation of the offline stage. We agree that adding an explicit evaluation metric after offline planning, and correlating it with specific online performance dimensions, would offer deeper insights into how well the blueprint is interpreted and whether emergent behaviors occur during enactment. We are currently conducting supplementary experiments to quantify this relationship and will include a detailed discussion on this correlation in the final version of the paper.



We hope these clarifications address some common concerns. We will incorporate these additional details into the final manuscript to strengthen the paper further.

---

### Author Response · Authors · 2025-11-26
**Request for Reviewer Follow-up on Rebuttal**

Dear Area Chairs, we have already submitted our rebuttal. We kindly ask that you remind any reviewers who have not yet responded to please take a moment to review our response and, if needed, provide follow-up comments—such as additional questions, requests for clarification, or updates to their original reviews.

We believe this will help ensure a thorough and fair evaluation of our work. We sincerely appreciate the thoughtful and insightful feedback from all four reviewers, which has significantly helped us improve the quality of our paper.

Thank you very much for your time and support.

---

### Author Response · Authors · 2025-12-01
**General Summary**

We sincerely thank the Area Chair and all reviewers as always, for their time and constructive engagement throughout the rebuttal period. We have uploaded a revised manuscript that incorporates the valuable feedback received.

For ease of tracking, all significant revisions in the PDF are highlighted in blue. We believe these improvements substantially strengthen the paper and address the concerns raised. We remain available for any further clarifications.

---

### Meta-Review · Area_Chair_9989 · 2026-01-08

**Summary:**

The initial reviews all indicated that the paper is interesting and the multi-agent framework as well as the setting is well motivated but had a few primary concerns: insufficient ablations regarding what parts of the multi-agent design worked or not, potential biases in the judge and lack of granularity in the dimensions of evaluation given a lack of human eval, and scalability for a live performance.

**Reviewer Concerns:**

The rebuttal response from the authors was quite extensive and seems to have addressed most of the reviewer's concerns. They added information regarding the actual multi-agent design and additional components require for live performances with detailed latency, etc information. They also added additional statistics for the evaluation e.g. Kripp's Alpha for the judge and increased the scope of the discussion in the paper. The remaining concerns are that the ablations to isolate components of the setup do not seem to have been performed to the reviewer's standards yet.

**Reviewer Scores:**

KH9F based on the back and forth and given that the reviewers responded directly to their points would likely raise their score to a 8

It seems likely that AUwa and ggC5 would increase their scores to a 6 given the depth of the rebuttal

bXsn would likely have kept their score as most of their concerns were rather cursory

---

### Decision · Program_Chairs · 2026-01-26

Accept (Poster)